# Data De-Duplication and Semantic Enhancement for Contrastive Language-Image Pre-training

## Abstract

Benefiting from the countless image-text pairs in the web data, vision-language pre-training models (*e.g.* CLIP) have emerged as an efficient alternative in learning representations that are transferable across a wide range of downstream tasks. However, we reveal that the web data are noisy, with significant scene redundancy and misalignment in the image-text pairs, which increase the training expenses and computing resources. To alleviate these problems, this paper proposes a novel training strategy that comprises two dedicated components, namely Data De-Duplication ($D^3$) and Semantic Enhancement (SE). $D^3$ leverages the pre-clustered data prototypes to decrease the training cost without reducing the data diversity by uniformly sampling a portion of image-text pairs at each training epoch. SE utilizes a large language model (LLM) and a visual large language model (VLLM) to refine and augment the text caption, which can help to form a one-to-multiple mapping relation between image and text. Furthermore, we employ a Diverse Captions Training Mechanism (DCTM) and a Modality Self-enhancement Training Mechanism (MSTM) for effective training. Experimental results indicate that the proposed method achieves state-of-the-art performance on various tasks including image classification, image-text retrieval, object detection, and segmentation (performance improvements varying from 0.2% to 23.9% for all datasets) with only half of the training time compared with original CLIP. Our code and generated data will be publicly available.

## 1 Introduction

Vision-language representation learning (Radford et al. (2021); Jia et al. (2021); Yuan et al.; Singh et al. (2022); Wang et al. (2022); Yu et al. (2022a); Sun et al. (2023)) has received increasing attention and achieved remarkable performance improvement in pre-training for many computer vision tasks with large-scale training data. Multimodal datasets are a critical component in recent breakthroughs such as CLIP and GPT4, but their design does not receive the same research attention as model architectures or training algorithms. The web data are noisy, with significant scene redundancy and misalignment in the image-text pairs, model training on these data is expensive and requires more computing resources.

The core idea is to construct a semantically diverse multimodal dataset and present an efficient training algorithm for better representation learning: Data De-Duplication ($D^3$) and Semantic Enhancement (SE) for Contrastive Language-Image Pre-training (DS-CLIP). Specifically, in $D^3$ as Figure 1(b), we first randomly select some images and feed them into an unsupervised visual encoder to extract visual features. Then, sampling is performed uniformly equiproportional based on the class center of pre-clustered data prototypes that ensures a balanced distribution of sampled data during each training epoch, substantially decreasing the training cost without reducing the scene diversity of data. For the SE module, we leverage powerful LLM (Touvron et al. (2023)) and VLLM (Liu et al. (2023)) in Figure 1(c) to enrich semantic information of text and mitigate the issue of text-image misalignment and enrich text descriptions.

Combining with refined and augmented texts, it forms a one-to-multiple mapping among image and text for the Diverse Captions Training Mechanism (DCTM) and Modality Self-enhancement

(a) Performance Comparison on Downstream Tasks    (b) Data De-Duplication Sampling    (c) Semantic Enhancement for Text

Figure 1: (a) Performance comparison on downstream tasks. (b) Proposed $D^3$ by clustering and sampling reduces training costs. (c) Proposed SE for text augmentation enriches textual information and alleviates image-text misalignment.

Training Mechanism (MSTM). In this way, DS-CLIP effectively reduces training time and alleviates data redundancy and misalignment by sampling only half of the augmented data during each training epoch. Training with less data on a semantically diverse dataset can achieve superior performance and save computing resources, which is important for studies with limited resources in the laboratory. The superiority of our method is illustrated in Figure 1(a).

To summarize, the main contributions of this work are listed as follows.

- We construct a semantically diverse multimodal dataset (the first publicly available high-quality image-text dataset with diverse semantic captions) for better representation learning through Semantic Enhancement (SE) modules, which effectively address the issue of image-text misalignment and enrich textual diversity.

- To effectively utilize the semantically diverse dataset, we design the Data De-Duplication ($D^3$), which manages a paradigm to reduce training costs without the loss of data diversity.

- With SE and $D^3$, we present an efficient training algorithm, called DS-CLIP using the Diverse Captions Training Mechanism (DCTM) and Modality Self-enhancement Training Mechanism (MSTM) to learn superior representation model. Our proposed method significantly outperforms CLIP on special fine-grained classification datasets and various patch-level downstream tasks from 0.2% to 23.9%, with ONLY half of the training time.

## 2   RELATED WORK

### 2.1   VISION-LANGUAGE REPRESENTATION LEARNING

Contrastive Language-Image Pre-training (CLIP) (Radford et al. (2021)) from a large-scale vision-language dataset has achieved significant progress in the computer vision community. Despite the impressive performance of CLIP, it is quite data-hungry and needs tens of millions or more image-text pairs for pre-training (Radford et al. (2021); Jia et al. (2021); Yuan et al.; Singh et al. (2022); Wang et al. (2022); Yu et al. (2022a); Sun et al. (2023)). However, obtaining large and curated multi-modality data is challenging due to the difficulties in annotation, which also leads to more expensive training efforts on the large-scale dataset.

To improve the efficiency of training, recent methods (Sorscher et al. (2022); Wang et al. (2023)) select only a subset of challenging samples and filter noisy data from the original dataset to pre-train the visual-language model. Some works (Mu et al. (2022); Li et al. (2022c)) introduce much-supervised information to help CLIP train better encoders on filtered data. These works achieve better performance on some datasets *e.g.*Imagenet (Deng et al. (2009)), but there is serious performance degradation on specific fine-grained datasets and patch-level downstream tasks because of the missing diversity in the condensed datasets. Other works integrate structured external data into the training stage to enhance feature representation and improve the zero-shot performance under the limitation of datasets instead of pursuing the large-scale dataset (Shen et al.; Li et al. (2022b)). These works achieve better performance but at the cost of increased computation. In this paper, we propose a novel and simple method to make full use of image-text pairs. Different from previous methods that focus on filtering noisy data to improve training efficiency or adding external data to enhance feature representation, we utilize half of the pre-clustered data with V/LLM-based aug-

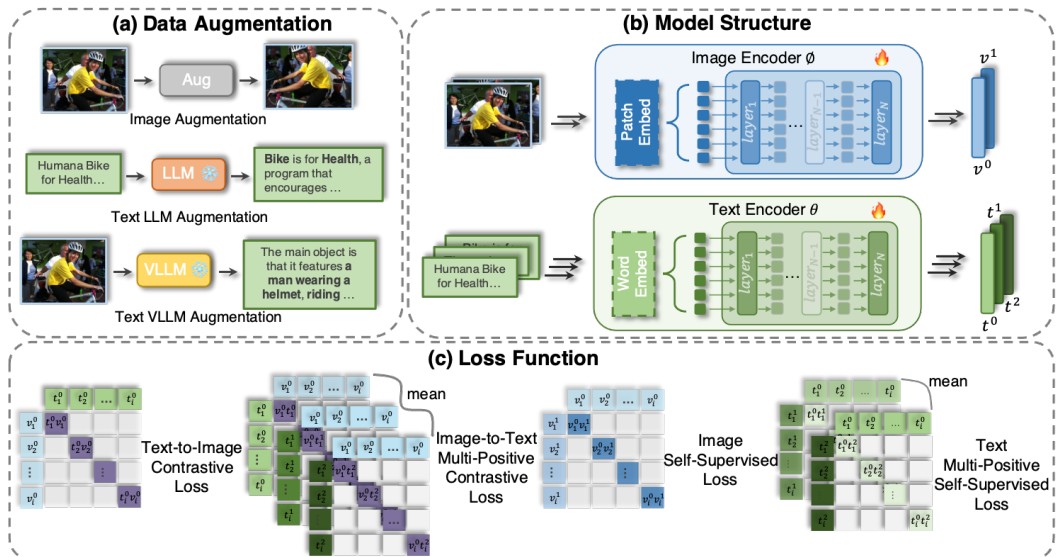

Figure 2: Framework of DS-CLIP. Our model is composed of (a) Data Augmentation, (b) Dual-Encoder Model Architecture, and (c) Loss Function. Data sampled from the pre-clustered prototype are augmented via image transformation and LLM and VLLM for texts. Then, corresponding features are extracted from the dual-encoder model architecture. The final loss is implemented via text-to-image contrastive loss, image-to-text multi-positive contrastive loss, self-supervised loss of images, and multi-positive self-supervised loss of texts.

mented texts during each training epoch, which achieves superior performance compared with CLIP trained on complete data.

## 2.2 TEXT AUGMENTATION FROM LARGE LANGUAGE MODEL

Natural language processing (NLP) task is another popular work and obtains remarkable performance improvement dependent on the quality and quantity of available data. Recently, models like the GPT-3 (Brown et al. (2020)), PaLM (Chowdhery et al. (2022)), instruction-GPT (Ouyang et al. (2022)), open-sourced LLaMA (Touvron et al. (2023)) and visual instruction model LLaVA (Liu et al. (2023)) achieve superior performance on various NLP tasks and visual tasks. With the success of LLM, recent works utilize LLM to improve performance for various visual-language understanding tasks. WaffleCLIP (Roth et al. (2023)) and DCLIP (Menon & Vondrick (2022)) employ LLM-generated descriptors to enrich the semantic representation of class (*e.g. A photo of a class, which is/has/etc a round shape, a grid pattern*). LaCLIP (Fan et al. (2023)) rewrites and changes the structure of text from the original text using LLaMA and chatbots but maintains semantic information of the text. These methods introduce various text representations but with similar semantic information, failing to resolve the misalignment issue. In this paper, we use LLaMA and LLaVA to rewrite text representations from the perspective of the original text and image. It enriches the text's expression from structure and semantics and alleviates the problem of image-text misalignment.

## 3 METHOD

In this section, we present the details of our proposed DS-CLIP. The framework of DS-CLIP for pre-training is summarized in Figure 2. DS-CLIP is a simple yet effective approach for vision-language representation learning, which not only achieves the reduction of the training cost but also enhances feature representation due to the enrichment of data diversity. Our approach consists of three stages: (1) Data De-Duplication ($D^3$) for pre-training data (Section 3.1), (2) Semantic Enhancement (SE) for text via LLM and VLLM (Section 3.2), (3) pre-training with Diverse Captions Training Mechanism (DCTM) and Modality Self-enhancement Training Mechanism (MSTM) (Section 3.3).

## 3.1 DATA DE-DUPLICATION FOR PRE-TRAINING

**Pre-clustered Data Prototypes.** Given $N$ image-text pairs $\{I_i, T_i\}_{i=1}^N$ as training data $\mathcal{T}$, where $I_i$ is the $i$-th image in the dataset and $T_i$ is its corresponding text description. Given $I_i$ as the

input image, we first feed it into an unsupervised pre-trained image encoder (*e.g.*, DINO (Caron et al. (2021))) to obtain the image embedding as $\mathbf{v}_i^c$. For $N$ encoded image features $\{I_i, T_i\}_{i=1}^N$, we randomly select $N'$ image features $\{\mathbf{v}_i^c\}_{i=1}^{N'}$ and use K-Means algorithm (Hartigan & Wong (1979)) to cluster with $K$ categories. Then, each image-text pair $\{I_i, T_i\}_{i=1}^N$ is marked as a clustered label based on the principle of the minimum Euclidean distance compared corresponding image feature with the feature of clustered center.

**Data De-Duplication for Pre-training.** We uniformly sample a certain percentage of data points from each pre-clustered center, producing a small subset of the dataset $\mathcal{T}_\mathcal{P}$ during each training epoch, which ensures the difference of sampled data and reduces data of scene duplicates. Different from the previous CLIP methods that only use $\{I_i, T_i\}_{i=1}^B$ to train encoders, we propose a novel utilization of the image-text pairs with multiple augmented texts introduced in Section 3.2. Specifically, each image has multiple corresponding rewritten texts, which forms a one-to-multiple mapping relation. In addition, each image is also augmented by transformations. By adopting generated texts and augmented images, we introduce a multi-positive contrastive training loss and self-supervised loss for pre-training, which is introduced in Section 3.3. The processing is shown in Figure 1(b).

## 3.2 SEMANTIC ENHANCEMENT FOR TEXT THROUGH LARGE LANGUAGE MODEL

**Text rewritten through LLaMA.** An important goal of vision-language representation learning is to align the embeddings of image and text (Radford et al. (2021)). We reveal that there are often several problems with the caption of the web image-text pairs, including grammar, spelling, semantic misalignment, etc. To guarantee the quality of the text data in the original web dataset, we propose to utilize the LLM to refine the caption of the image-text pair. Inspired by (Fan et al. (2023)), we propose leveraging LLM (*e.g.* LLaMA (Touvron et al. (2023))) and VLLM (*e.g.* LLaVA (Liu et al. (2023))) to generate more semantic texts to eliminate the above problem. The detail of the text rewritten is shown in Appendix.A. Refined text through LLaMA is represented as $T_i^1$. The refined captions keep the essence of the original captions with improved and diversified style and details, which enrich the text representation.

**Text augmentation through LLaVA.** Text rewritten through LLaMA only ensures the semantics associated with the corresponding original text. However, we reveal that large-scale datasets usually contain many noises and misaligned image-text pairs. Text augmentation through LLaMA alone can not alleviate the misalignment problem of image-text pairs. LLaVA (Liu et al. (2023)) is a visual instruction model that generates text descriptions from images. Its performance is comparable to human-level on various visual tasks. Therefore, we utilize LLaVA to generate texts from the perspective of the image. Details of generated texts from LLaVA are shown in Appendix.A. The generated text through LLaVA is $T_i^2$. By incorporating text augmentation methods LLaMA and LLaVA, we create a comprehensive guiding method in effectively generating diverse semantically consistent and modality-aligned texts as shown in Figure 1(c).

## 3.3 OVERALL TRAINING FRAMEWORK

**Dual-Encoder Architecture.** The dual-encoder architecture in DS-CLIP consists of an image encoder and a text encoder similar to the original CLIP (Radford et al. (2021)). The image encoder used in our settings is a Vision Transformer (Dosovitskiy et al. (2020)). Given an image $I_i$, we take the embedding of the $\ell$-2 normalized `[CLS]` token to obtain the representation of $I_i$, which is a $d$-dimensional feature vector $\mathbf{v}_i^0$. The text encoder is also a Transformer (Dosovitskiy et al. (2020)), which transforms and normalizes a sequence of input text $T_i$ into another $d$-dim feature vector $\mathbf{t}_i^0$.

**Diverse Captions Training Mechanism.** In this paper, following the standard method CLIP (Radford et al. (2021)) and the unified contrastive learning method (Yang et al. (2022)), we adopt the extension of transforming image-text pairs from one-to-one mapping to one-to-multiple mapping relation, as the description of text augmentation in Section 3.2.

Specifically, given a batch of $B$ image-text pairs $\{I_i, T_i\}_{i=1}^B$ as training data, DS-CLIP produces the augmented texts from LLaMA and LLaVA respectively: $\{T_i^1\}_{i=1}^B$ and $\{T_i^2\}_{i=1}^B$, where each input image $I_i$ has a set of corresponding diverse rewritten texts $\{T_i, T_i^1, T_i^2\} \in \mathcal{R}(i)$, which forms the one-to-multiple relation between image and text. DCTM employs text-to-image contrastive loss

$\mathcal{L}_{t2v}$ and image-to-text multi-positive contrastive loss $\mathcal{L}_{v2t}$ to align the image embedding $\mathbf{v}_i^0$ with a set of text embeddings $\mathbf{t}_i^m$, as shown in equation 1 and equation 2, it calculates the mean of the contrastive loss for the image to each corresponding text and standard contrastive loss for the original text to the original image.

$$\mathcal{L}_{v2t} = -\frac{1}{M+1} \sum_{m=0}^{M} \log \left( \frac{\exp(\sigma(\mathbf{v}_i^0, \mathbf{t}_i^m)/\tau)}{\sum_{j=1}^{B} \exp(\sigma(\mathbf{v}_i^0, \mathbf{t}_j^m)/\tau)} \right). \tag{1}$$

$$\mathcal{L}_{t2v} = -\log \left( \frac{\exp(\sigma(\mathbf{t}_i^0, \mathbf{v}_i^0)/\tau)}{\sum_{j=1}^{B} \exp(\sigma(\mathbf{t}_i^0, \mathbf{v}_j^0)/\tau)} \right). \tag{2}$$

where $M = 2$. $\mathbf{t}_i^m$ is the $i$-th group feature of augmented texts, $\mathbf{t}_i^m \in \left\{ \mathbf{t}_i^0, \mathbf{t}_i^1, \mathbf{t}_i^2 \right\}$. $\mathbf{t}_i^1$ is the feature of refined text from LLaMA and $\mathbf{t}_i^2$ is the feature of generated text from LLaVA. $\tau$ is a learnable temperature parameter to scale the pairwise cosine similarities, which is initialized as 0.07. $\sigma(\cdot, \cdot)$ computes the cosine similarity between two vectors.

**Modality Self-enhancement Training Mechanism.** In addition, we leverage self-supervision within each modality for self-enhancement. Images are also augmented through transformation (*e.g.* adding noise, interpolation, random cropping) as $\left\{ I_i^1 \right\}_{i=1}^{B}$. With the diverse texts from SE, MSTM leverages the self-supervised loss of image $\mathcal{L}_{v2v}$ and multi-positive self-supervised loss of text $\mathcal{L}_{t2t}$ within each modality, as shown in equation 3 and equation 4, it calculates the mean of the self-supervision loss for the multiple text pairs, and standard self-supervision loss for the image and its augmented image.

$$\mathcal{L}_{v2v} = -\log \left( \frac{\exp(\sigma(\mathbf{v}_i^0, \mathbf{v}_i^1)/\tau)}{\sum_{j=1}^{B} \exp(\sigma(\mathbf{v}_i^0, \mathbf{v}_j^1)/\tau)} \right). \tag{3}$$

$$\mathcal{L}_{t2t} = -\frac{1}{M} \sum_{m=1}^{M} \log \left( \frac{\exp(\sigma(\mathbf{t}_i^0, \mathbf{t}_i^m)/\tau)}{\sum_{j=1}^{B} \exp(\sigma(\mathbf{t}_i^0, \mathbf{t}_j^m)/\tau)} \right). \tag{4}$$

Overall, the total loss of our framework is as follows.

$$\mathcal{L} = \mathcal{L}_{v2t} + \mathcal{L}_{t2v} + \alpha \mathcal{L}_{v2v} + \beta \mathcal{L}_{t2t}. \tag{5}$$

where $\alpha$ and $\beta$ are hyperparameters.

## 4 EXPERIMENTS

In this section, we first describe the implementation details in Section 4.1. Then we conduct ablation experiments in Section 4.2 to validate the effectiveness of the proposed method and analyze the effect of each component. Finally, we compare our method with recently proposed relevant CLIP-variants in Section 4.3 and discuss the superiority of our DS-CLIP.

### 4.1 IMPLEMENTATION DETAILS

**Model Architecture.** The proposed framework is a general dual-encoder architecture, which consists of an image encoder and a text encoder similar to the original CLIP (Radford et al. (2021)). By default, the image encoder is a random initialized ViT-B/32 (Dosovitskiy et al. (2020)) and the text encoder is a BERT-base (Devlin et al. (2019)) model. Other details of the model architecture are shown in Appendix.B.

**Dataset and Downstream Tasks.** All the experiments are conducted on publicly available datasets. For a fair comparison, we follow previous works (Radford et al. (2021); Mu et al. (2022)) and use a 15 million English subset of YFCC (Thomee et al.) for pre-training on the ViT-B model. Another dataset such as Laion400M (Schuhmann et al.) is also used for pre-training the large backbone. After pre-training, the proposed model is evaluated in a zero-shot setting on several downstream

datasets. For the image classification task, 10 wide visual recognition datasets are used to evaluate the zero-shot classification performance and linear probe classification performance, including ImageNet (Deng et al. (2009)), ImageNetV2 (Recht et al. (2019)), CIFAR10 (Krizhevsky et al. (2009)), CIFAR100 (Krizhevsky (2009)), Caltech101 (Fei-Fei et al. (2005)), Oxford Pets (Parkhi et al.), SUN397 (Xiao et al. (2014)), Food101 (Bossard et al. (2014)), DTD (Cimpoi et al. (2013)) and Stanford Dogs (Khosla et al. (2011)). Top-1 accuracy is used for classification evaluation. For the zero-shot retrieval task, we perform the evaluation on MS-COCO (Chen et al. (2015)) and Flickr30K (Young et al. (2014)) for image-text retrieval and MSRVTT (Xu et al. (2016)) for video-text retrieval. We use the R@K to report the recall of top-K retrieval items. Following MIMdet (Fang et al. (2022)) and MMsegmentation (Contributors (2020)), two datasets are used to evaluate the performance on downstream tasks, i.e., COCO (Chen et al. (2015)) and ADE20K (Zhou et al. (2019)). The mean Average Precision (mAP) and mean Intersection over Union (mIoU) are used to evaluate performance. For the image captioning task, we consider the COCO dataset with BLEU@4 and CIDEr indicator following coca (Yu et al. (2022b)) for a fair comparison. More details about the datasets can be found in Appendix.B.

**Training Settings.** Details of training settings are shown in Appendix.B.

## 4.2 ABLATION STUDY

In this section, we demonstrate the effectiveness of $D^3$ in reducing training costs without compromising performance and validate the ability of SE to diversify the data and improve performance. Moreover, we further explore the efficacy of our proposed DCTM and MSTM. Furthermore, we discuss the impact of visual language models introduced in Section 3.2, which generate visual text descriptions from images, on the model's performance. Finally, we show the performance of our model on a large dataset using a large model. More experiments with different designs of $D^3$ and SE are provided in Appendix.D. Table 1 shows the performance of DS-CLIP on ImageNet classification with different proposed components. We test different backbones and datasets including training ViT-B/32 and ViT-B/16 on YFCC15M and training ViT-L/14 on Laion400M. Table 2 shows the performance of our DS-CLIP on zero-shot ImageNet classification with different design choices of $D^3$ and SE trained with ViT-B/32 on YFCC15M.

**$D^3$ effectively reduces training costs and maintains performance.** Table 2b reveals that utilization of $D^3$ enables models trained with only 70% of the training samples to achieve comparable performance to CLIP trained on the entire dataset. Notably, when employing a 50% sampling ratio, the model trained with pre-clustered prototype sampling outperforms the model trained with random sampling, as indicated by Experiment ID 2 and 3 in Table 1. Additional experiments involving different sampling ratios are compared in Appendix.D. In addition, we ablate the clustering feature type and number, sampling type and number in pre-training are shown in Appendix.D. The superiority of the pre-clustered prototype sampling can be attributed to the preservation of sample diversity during each training epoch.

**SE significantly enriches texts and addresses the issue of text-image misalignment.** Text augmentation from LLM and VLLM is the key factor in our proposed DS-CLIP. From Table 1, we observe that with the help of the text augmentation, the proposed method outperforms the baseline with +3.3% (experiment ID 4 vs. experiment ID 3), +4%(experiment ID 7 vs. experiment ID 1), which is trained on clustered samples of 50% and all samples respectively. By training the model with only 50% augmented clustered data, we were able to achieve comparable performance to that of CLIP, which used 100% of the available data. Furthermore, our model shows performance improvement when trained with clustered data sampled at different ratios, demonstrating the effectiveness of the SE module, as shown in Table 2a. The generated texts provide rich and semantic information for each original text, which would boost the model to learn a more robust representation for image classification. From Table 2c, experiments are also performed separately for the two ways of text augmentation. They are all beneficial for vision-language learning and text enhancement from LLaVA is more effective than LLaMA, for text augmentation from *e.g.* LLaVA relegates the misalignment problem and the generated texts are more robust. More visual results are shown in Appendix.E. Training with generative semantic data enriches the representation of text, it is verified that is beneficial to vision-language learning through the above experiments.

Table 1: Ablation experiments on zero-shot ImageNet classification with our proposed components. All 100%, Random 50%, and Clustered 50% represent training with all data, randomly selecting 50% data for training and sampling 50% data according to the pre-clustered prototype during the training epoch. "–" is no training. The default setting of hyper-parameter clustered number $K$ is equal to 10000 using the image feature for clustering and uniform sampling type.

| ID | Method | Init. of Image Enc. | $D^3$ | SE/ LLaVA | SE/ LLaMA | DCTM | MSTM | ImageNet Top-1 |
|----|--------|---------------------|-------|-----------|-----------|------|------|----------------|
| \multicolumn{9}{c}{Dataset: YFCC15M; Model Architecture: ViT-B/32} | | | | | | | | |
| 1 | CLIP | ViT rand. | All 100% | ✗ | ✗ | ✗ | ✗ | 37.7 |
| 2 | CLIP | ViT rand. | Random 50% | ✗ | ✗ | ✗ | ✗ | 32.5 |
| 3 | DS-CLIP | ViT rand. | Clustered 50% | ✗ | ✗ | ✗ | ✗ | 34.6 |
| 4 | DS-CLIP | ViT rand. | Clustered 50% | ✓ | ✓ | ✗ | ✗ | 37.9 |
| 5 | DS-CLIP | ViT rand. | Clustered 50% | ✓ | ✓ | ✓ | ✗ | 38.5 |
| 6 | DS-CLIP | ViT rand. | Clustered 50% | ✓ | ✓ | ✓ | ✓ | 39.3 |
| 7 | DS-CLIP | ViT rand. | All 100% | ✓ | ✓ | ✗ | ✗ | 41.7 |
| 8 | DS-CLIP | ViT rand. | All 100% | ✓ | ✓ | ✓ | ✓ | **42.1** |
| \multicolumn{9}{c}{Dataset: YFCC15M; Model Architecture: ViT-B/16} | | | | | | | | |
| 9 | CLIP | ViT rand. | All 100% | ✗ | ✗ | ✗ | ✗ | 43.4 |
| 10 | CLIP | BLIP init. | – | ✗ | ✗ | ✗ | ✗ | 43.8 |
| 11 | CLIP | BLIP init. | All 100% | ✗ | ✗ | ✗ | ✗ | 44.3 |
| 12 | DS-CLIP | ViT rand. | All 100% | ✓ | ✗ | ✗ | ✗ | 45.7 |
| 13 | DS-CLIP | ViT rand. | Clustered 50% | ✓ | ✓ | ✓ | ✓ | 45.4 |
| 14 | DS-CLIP | ViT rand. | All 100% | ✓ | ✓ | ✓ | ✓ | **47.7** |
| \multicolumn{9}{c}{Dataset: Laion400M; Model Architecture: ViT-L/14} | | | | | | | | |
| 15 | CLIP | ViT rand. | All 100% | ✗ | ✗ | ✗ | ✗ | 75.0 |
| 16 | CLIP | LLaVA init. | – | ✗ | ✗ | ✗ | ✗ | 75.3 |
| 17 | CLIP | LLaVA init. | All 100% | ✗ | ✗ | ✗ | ✗ | 76.0 |
| 18 | DS-CLIP | ViT rand. | All 100% | ✓ | ✗ | ✗ | ✗ | 76.7 |
| 19 | DS-CLIP | ViT rand. | Clustered 50% | ✓ | ✓ | ✓ | ✓ | 76.1 |
| 20 | DS-CLIP | ViT rand. | All 100% | ✓ | ✓ | ✓ | ✓ | **78.7** |

**DCTM and MSTM can significantly enhance uni-modal and cross-modal alignment.** DCTM leverages multi-positive contrastive loss in each image with multiple diverse texts to enhance the alignment of image-text embeddings. To explore the effectiveness of DCTM, we compare the performance of DCTM with only using single text during pre-training. Specifically, in the single text setting, only one text is sampled from multiple generating texts with the standard text-image contrastive loss (Radford et al. (2021)), as Experiment ID 4 in Table 1. Experiment ID 4-5 in Table 1 shows that multiple text embeddings for DCTM bring +0.6% improvement in the accuracy of classification, from which we can indicate that training with DCTM could implicitly help the model to improve the data diversity and thus align the image and text embeddings more efficaciously.

To further effectively improve the uni-modal representational learning, we leverage the self-supervision within each visual and language modality to enhance the representation of image and text embeddings, called MSTM as Section 3.3. Experiment ID 6 in Table 1 shows the efficacy of MSTM. In sum, we can draw the conclusion that DCTM and MSTM could jointly help to exploit the potential information of data and significantly enhance uni-modal and cross-modal alignment.

**Generated text from LLaVA is more helpful than training initialization based on LLaVA.** To verify the superior performance of DS-CLIP is coming from augmented texts instead of these pre-trained models, we employ the equivalent parameters image encoder of BLIP and the large image encoder of LLaVA to initialize the image encoder ViT-B/16 and ViT-L/14 of DS-CLIP respectively, and other parameters are kept maintaining. We evaluated the direct results of these pre-trained models (Experiment ID 10 and 16) and further trained results on the YFCC15M and Laion400M datasets (Experiment ID 11 and 17). Experimental results show that using the pre-trained image encoder of the BLIP and LLaVA to initialize the image encoder of DS-CLIP, experiments ID( 10, 11, 16 and 17) provide promising results in Table 1. Furthermore, results in training with augmented texts from BLIP (Li et al. (2022a)) and LLaVA (Liu et al. (2023)), *e.g.* Experiment ID 12 and 18, bring higher 1.4%, 0.7% compared to model training with initialization from image encoder of BLIP (Experiment ID 11) and LLaVA (Experiment ID 17). These experiments show that the performance of our model is not limited by the pre-trained model like LLaVA.

**DS-CLIP is scalable for training on large datasets and models.** To evaluate the scalability of the proposed framework, we train the proposed model on a larger dataset Laion400M with a large ViT-L/14 encoder following the settings in CLIP. The results in Table 1 show that the proposed method outperforms the CLIP baseline on ImageNet classification when pre-training on the large-scale dataset. We also verify that DS-CLIP is effective irrespective of the underlying backbone architecture (Experiment ID 6, 13, 19) in Table 1. In addition, it takes about 175 hours to rewrite one text from LLaMA for the entire Laion400M dataset on 32 A100 GPU machines, and it takes only 75 hours to generate texts from LLaVA with batch inference. The generated texts will be released.

Table 2: Ablation studies on different design choices of $D^3$ and SE. All models are trained on YFCC15M and evaluated on zero-shot ImageNet classification.

(a) Ablating SE on multi-scale sample data.

| Number | SE | Top-1 | Number | SE | Top-1 |
|--------|-----|-------|--------|-----|-------|
| 10%    | ✗  | 18.3  | 50%    | ✗  | 34.6  |
|        | ✓  | 22.5  |        | ✓  | 37.9  |
| 30%    | ✗  | 28.6  | 70%    | ✗  | 37.6  |
|        | ✓  | 30.4  |        | ✓  | 40.2  |

(b) Sample number.

| Sample Number | Top-1 |
|---------------|-------|
| 10%           | 18.3  |
| 30%           | 28.6  |
| 50%           | 34.6  |
| 70%           | 37.6  |

(c) Text augmentation.

| LLaVA | LLaMA | Top-1 |
|-------|-------|-------|
| ✗    | ✗    | 34.6  |
| ✗    | ✓    | 36.5  |
| ✓    | ✗    | 37.6  |
| ✓    | ✓    | 37.9  |

## 4.3 MAIN RESULTS

Table 3: Zero-shot image classification performance and linear probe classification performance on 10 downstream datasets(%).

| Evaluation Type | Method | IN | INV2 | Pets | C10 | C100 | S397 | F101 | Cal101 | DTD | Dogs | Avg. |
|-----------------|--------|-----|------|------|------|------|------|------|--------|------|------|------|
| Zero-shot | CLIP Radford et al. (2021) | 37.7 | 32.8 | 16.1 | 76.0 | 48.6 | 50.8 | 21.8 | 69.8 | 28.3 | 11.1 | 39.3 |
|  | MS-CLIP You et al. (2022) | 36.7 | 30.2 | – | – | – | – | – | – | – | 5.6 | – |
|  | SLIP Mu et al. (2022) | 38.3 | 33.3 | 28.3 | 72.2 | 45.3 | 45.1 | 44.7 | 65.9 | 21.8 | 11.8 | 40.7 |
|  | DeCLIP Li et al. (2022c) | 43.2 | 36.1 | 30.2 | 72.1 | 39.7 | 51.6 | 46.9 | 70.1 | 24.2 | 11.7 | 42.6 |
|  | LaCLIP Fan et al. (2023) | 38.1 | 34.2 | 32.1 | 77.5 | 50.3 | 47.2 | 45.8 | 79.8 | 27.5 | 10.2 | 44.3 |
|  | DS-CLIP | 39.3 | 34.4 | 32.4 | 78.1 | 53.7 | 50.8 | 45.7 | 83.9 | 29.0 | 12.3 | **45.9** |
| Linear probe | CLIP Radford et al. (2021) | 63.5 | 51.3 | 69.8 | 91.7 | 74.1 | 64.7 | 69.1 | 84.9 | 66.5 | 50.5 | 68.6 |
|  | MS-CLIP You et al. (2022) | 68.1 | 49.8 | 62.1 | 87.2 | 66.7 | 71.7 | 76.0 | 81.6 | 69.4 | 46.1 | 67.9 |
|  | SLIP Mu et al. (2022) | 68.1 | 52.1 | 75.4 | 90.5 | 75.3 | 73.5 | 77.1 | 87.2 | 71.1 | 52.6 | 72.3 |
|  | DeCLIP Li et al. (2022c) | 69.2 | 53.1 | 76.5 | 88.6 | 71.6 | 75.9 | 79.3 | 88.0 | 69.1 | 49.9 | 72.1 |
|  | LaCLIP Fan et al. (2023) | 71.1 | 54.5 | 73.1 | 91.0 | 75.6 | 73.1 | 75.8 | 89.9 | 73.8 | 53.4 | 73.1 |
|  | DS-CLIP | 71.1 | 55.2 | 71.3 | 91.9 | 74.9 | 75.2 | 77.3 | 92.0 | 74.9 | 65.8 | **74.9** |

In this section, we compare DS-CLIP with CLIP and recently proposed relevant CLIP-variant (*e.g.* CLIP with self-supervised (SLIP), Modality-Shared CLIP (MS-CLIP), Data efficient CLIP (De-CLIP), and language augmentation CLIP (LaCLIP)) on two downstream visual recognition tasks: zero-shot image classification and linear probe image classification. Evaluations are conducted on 10 widely used visual recognition benchmarks including ImageNet (IN), ImageNetV2 (INV2), CI-FAR10 (C10), CIFAR100 (C100), Caltech101 (Cal101), Oxford Pets, SUN397 (S397), Food101 (F101), DTD and Stanford Dogs. In order to verify the representation learning ability of the cross-modal, we also test DS-CLIP on the image-text retrieval task on COCO (Chen et al. (2015)) and Filckr30K (Young et al. (2014)), and video-text retrieval task on MSRVTT (Xu et al. (2016)). Besides that, following MIMdet (Fang et al. (2022)) and MMSegmentation (Contributors (2020)), we finetune the image encoder of DS-CLIP to evaluate the performance on COCO (Chen et al. (2015)) and ADE20K (Zhou et al. (2019)). Furthermore, we also conducted the image captioning task on the COCO dataset (Chen et al. (2015)). Previous methods usually take YFCC15M or its augmented version as a training set. By default, we sample 50% data during each epoch training according to clustered prototypes, with each sample augmented and trained with DCTM and MSTM of DS-CLIP in Table 3 as Experiment ID 6 in Table 1.

**Zero-shot image classification.** We report the performance of DS-CLIP on all 10 datasets as well as the results of previous methods CLIP, SLIP, MS-CLIP, DeCLIP, and LaCLIP. DS-CLIP shows superior performance than previous state-of-the-art methods. In particular, the performance of DS-CLIP significantly outperforms other methods on the fine-grained dataset, which brings +13.3% improvement on dogs and 8.2% increase on Food101. The generated texts could provide more fine-grained

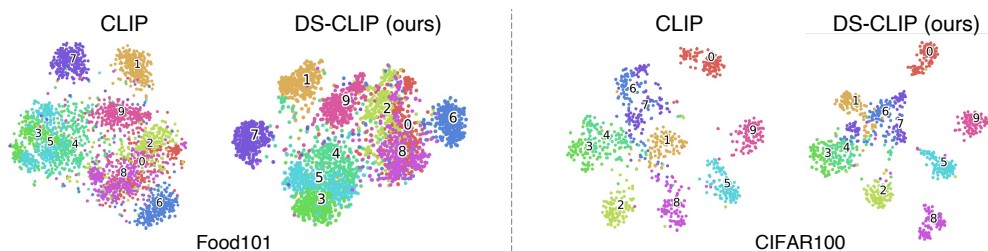

Figure 3: Distribution of visual embedding space with CLIP and DS-CLIP using t-SNE.

Table 4: Performance of DS-CLIP on downstream tasks(detection, segmentation and retrieval) (%).

| Method | Detection | Segmentation | | Image-Text Retrieval | | | | Video-Text Retrieval | | Image Caption | |
|---|---|---|---|---|---|---|---|---|---|---|---|
| | COCO | COCO | ADE20k | COCO | | Flickr30K | | MSRVTT | | COCO | |
| | mAP | mAP | mIOU | I2T/R@1 | T2I/R@1 | I2T/R@1 | T2I/R@1 | V2T/R@1 | T2V/R@1 | BLEU@4 | CIDEr |
| CLIP | 44.8 | 40.1 | 46.0 | 20.4 | 33.3 | 37.8 | 57.3 | 9.1 | 9.1 | 33.7 | 108.3 |
| DS-CLIP | 46.9 (+2.1) | 41.1 (+1.0) | 48.3 (+2.3) | 30.7 (+10.3) | 45.5 (+12.2) | 52.3 (+14.5) | 75.3 (+18.0) | 15.1 (+6.0) | 15.1 (+6.0) | 35.4 (+1.7) | 112.5 (+4.2) |

information for learning a better image encoder. Furthermore, we also provide the distribution of visual embedding space with CLIP and DS-CLIP on Food101 and CIFAR100 using t-SNE (Maaten & Hinton (2008)) in Figure 3, from which we can indicate that diverse textual information helps learn a better image encoder to distinguish the boundaries of each class. More visualization of 10 classification datasets is shown in Appendix.F.

**Linear probe image classification.** Following previous methods, we also conduct linear probe image classification experiments to test the transferability of the learned image representation. We fix the pre-trained image encoder of our trained DS-CLIP, then train a linear classifier to conduct image classification on downstream datasets. Experimental results are shown in Table 3, where DS-CLIP achieves better performance than previous methods CLIP, MS-CLIP, SLIP, DeCLIP, and LaCLIP, it brings 6.3% (74.9% v.s. 68.6%) averaged improvements compared with baseline.

**Performance on downstream tasks.** For downstream tasks, we use pre-trained VIT-B/16 as Experiment ID 13 in Table 1 to initialize the image encoder of CLIP for fair comparison following the original code of MIMdet and MMSegmentation, and other parameters are maintained. **For the dense prediction tasks,** one can tell that the recognition ability on the patch level is improved with generated texts for detection and segmentation tasks, as Table 4. Specifically, DS-CLIP still outperforms CLIP on these two benchmarks with less training data, improving 2.1% on COCO for detection, 1.0% on COCO, and 2.3% on ADE20K for segmentation, respectively. We conjecture that sampling from generative texts could provide richer and more fine-grained information for better dense prediction tasks. **For the image-text retrieval task,** significant improvements on COCO and Flickr30K are achieved when learning with generated embedding, as shown in Table 4. Specifically, the proposed method surpasses the baseline with +10.3%, +12.2%, +14.5%, +18.0% on image-to-text and text-to-image retrieval top-1 accuracy of COCO and Flickr30K, respectively. Training with more semantic generative text could provide implicit and rich information that forces the model to align the image and text embeddings, which is beneficial to the cross-modal downstream tasks. **For video-text retrieval tsak**, zero-shot DS-CLIP outperforms baseline CLIP as shown in Table 4. **For the image captioning task**, we use image encoder VIT-B/16 of CLIP and DS-CLIP to initialize the image encoder of coca (Yu et al. (2022b)) for fair comparison following the original code training on the COCO caption trainset. The result is shown in Table 4, we can find out DS-CLIP far exceeds baseline CLIP by 1.7% of BLEU@4 and 4.2 of CIDEr.

## 5 CONCLUSION

In this paper, we present DS-CLIP, a novel and efficient method that maximizes the benefits of image-text pairs, achieving reduced training costs and increased dataset diversity. Unlike previous CLIP approaches, which solely rely on image-text pairs for encoder training, DS-CLIP incorporates the $D^3$ and SE modules, where $D^3$ enables data de-duplication by clustering data and sampling image-text pairs, while SE addresses the issue of image-text misalignment by leveraging LLM and VLLM for generating semantic and fine-grained text descriptions. Experimental results demonstrate the effectiveness of DS-CLIP across various tasks. We hope that our research inspires further exploration in the realm of data-efficient utilization in CLIP.

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
