# DATA DE-DUPLICATION AND SEMANTIC ENHANCEMENT FOR CONTRASTIVE LANGUAGE-IMAGE PRE-TRAINING-SUPPLEMENTARY

This appendix is organized as follows.

- In Section A, we describe the processing of our text augmentation.

- In Section B, we introduce implementation details of the our experiment.

- In Section C, we list the prompt templates used for zero-shot classification experiments.

- In Section D, we provide more experiments to explore components of DS-CLIP.

- In Section E, we offer more visualization results of augmented text for DS-CLIP.

- In Section F, we show the distribution of visual embedding space with CLIP and DS-CLIP on 10 classification datasets.

## A  DETAILS OF TEXT AUGMENTATION

Figure 1 shows processing of text augmentation from LLaMA (Touvron et al. (2023)) and LLaVA (Liu et al. (2023)).

For LLaMA-based Text2Text rewritten, firstly, following LaCLIP (Fan et al. (2023)), we randomly select some texts from pre-training image-text datasets and then generate modified text through ChatGPT (Sardana et al.) as follows. A prompt "Rewrite this caption in detail and no more than 77 words." is defined. The selected texts are then inputted into ChatGPT with the prompt to generate a refined caption. Secondly, we randomly sampled five origin-rewrite text pairs as instances. Given a text sample to be processed, the instances, the prompt "Rewrite this caption in details and no more than 77 words." and an especial symbol "==>" are together inputted into LLaMA to generate refined text.

For LLaVA-based Image2Text generating, the processing is shown as follows. Given an image sample to be generated text, firstly, we design a sentence prompt that informs LLaVA about the task of generating image descriptions, such as "Tell me in detail what is main object description in this picture.". This serves as an initial contextual clue for the LLaVa to understand the objective at hand. Then, the prompt combined with the corresponding image further enables the LLaVA model to generate more semantic texts.

## B  IMPLEMENTATION DETAILS

**Model Architecture.** The proposed framework is a general dual-encoder architecture, which consists of an image encoder and a text encoder similar to the origin CLIP (Radford et al. (2021)). By default, the image encoder is a random initialized ViT-B/32 (Dosovitskiy et al. (2020)), which has 12 layers of transformer blocks. Each transformer block has 12 attention heads and the hidden size is set to 768. The text encoder is BERT-base (Devlin et al. (2019)), which shares a similar scale with the image encoder, *e.g.* 12 layers of transformer blocks, 12 attention heads in each transformer block, and 768 hidden sizes. The output of the text encoder and image encoder is projected to 384-dim by linear projection. We adopt the unsupervised model DINO-S/8 (Caron et al. (2021)) as the image encoder to pre-compute image features for clustering. We truncate the input text tokens so that they have a maximum length of 77. The input image is resized to $224 \times 224$, and results in $7 \times 7$ image patches when the patch size is $32 \times 32$ for ViT-B/32.

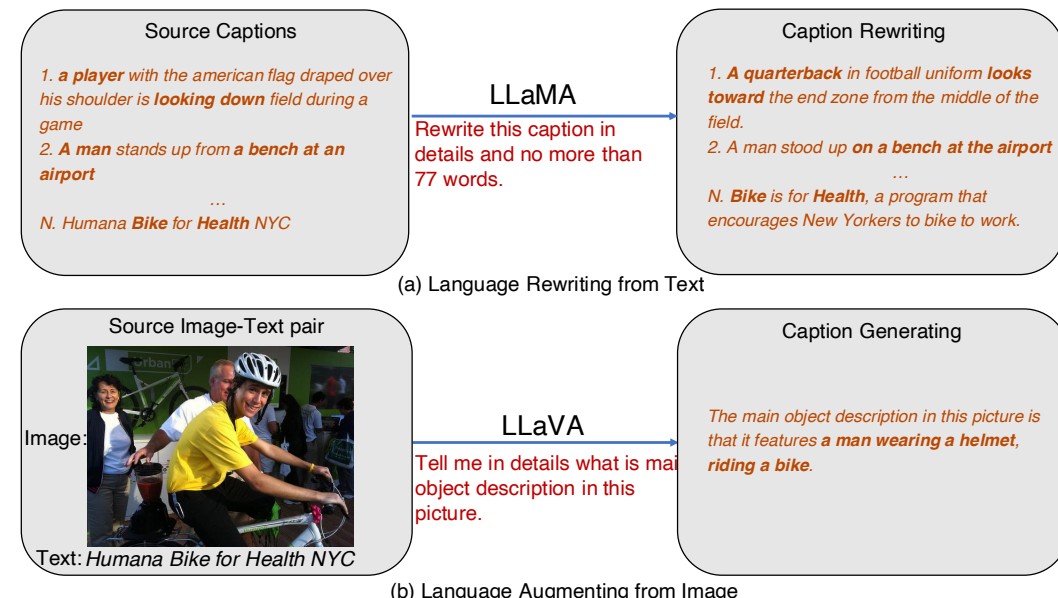

Figure 1: Text Augmentation through LLM and VLLM.

**Dataset and Downstream Tasks.** We have 13 widely used downstream datasets: ImageNet (Deng et al. (2009)), ImageNetV2 (Recht et al. (2019)), CIFAR10 (Krizhevsky et al. (2009)), CIFAR100 (Krizhevsky (2009)), Caltech101 (Fei-Fei et al. (2005)), Oxford Pets (Parkhi et al.), SUN397 (Xiao et al. (2014)), Food101 (Bossard et al. (2014)), DTD (Cimpoi et al. (2013)), Stanford Dogs (Khosla et al. (2011)), COCO (Chen et al. (2015)), ADE20K (Zhou et al. (2019)), Flickr30K (Young et al. (2014)) and MSRVTT (Xu et al. (2016)). Table 1 summarizes the details of these datasets. For ImageNetV2, we use the same training data of ImageNet for the linear probe classification experiments. For COCO, Flicker30K, and MSRVTT for the image-text and video-text retrieval task, we only evaluate the zero-shot top-K retrieval items, thus we don't need training data. The pre-trained model is used to extract embeddings from images, videos, and texts separately. Similarity scores between image/video embeddings and text embeddings are used for ranking. We use the R@K to report the recall of top-K retrieval items. There are 30,000 images and each image has corresponding five text on Flickr30K. COCO and ADE20K are used for detection and segmentation evaluation. The classification datasets use classification accuracy as an evaluation metric, except for Caltech101 and Oxford Pets, which use averaged per-class accuracy. The detection datasets use mean average precision as the evaluation metric. The segmentation datasets use mean Intersection over Union(mIoU) as an evaluation metric.

Table 1: Details of downstream datasets.

| Dataset | #Classes | #Train | #Test | Metric |
|---|---|---|---|---|
| ImageNet | 1,000 | 1,281,167 | 50,000 | accuracy |
| ImageNetV2 | 1,000 | – | 50,000 | accuracy |
| CIFAR10 | 10 | 50,000 | 10,000 | accuracy |
| CIFAR100 | 100 | 50,000 | 10,000 | accuracy |
| Caltech101 | 102 | 3,060 | 6,085 | mean-per-class |
| Oxford Pets | 37 | 3,680 | 3,669 | mean-per-class |
| SUN397 | 397 | 19,850 | 19,850 | accuracy |
| Food 101 | 102 | 75,750 | 25,250 | accuracy |
| DTD | 47 | 3,760 | 1,880 | accuracy |
| Stanford Dogs | 120 | 12,000 | 8,580 | accuracy |
| COCO | 81 | 118287 | 5,000 | mean average precision |
| ADE20K | 150 | 25574 | 2000 | mean Intersection over Union |
| Filcker30K | – | – | 30000 | recall of top-K retrieval item |
| MSRVTT | – | – | 1000 | recall of top-K retrieval item |

**Training Settings.** We implement our framework with Pytorch (Paszke et al. (2019)). All pre-training experiments with ViT-B are conducted on 8 NVIDIA Tesla A100 GPUs, and experiments with ViT-L are conducted on 96 NVIDIA Tesla A100 GPUs. The batch size is set to 4096. The framework is trained for 32 epochs with LAMB optimizer (You et al. (2020)) and an initial learning rate of 2.5e-3. The learning rate follows a cosine decay schedule with 5 epochs of linear warm-up. Weight decay is set to 0.2. For data augmentation, we random crop a $224{\times}224$ patch from the input image, then conduct random horizontal flip, random color distortions, random Gaussian blur, and RandAugment (Cubuk et al. (2020)), following previous works (Mu et al. (2022); You et al. (2022); Radford et al. (2021)). $\alpha$ and $\beta$ are equal to 1 in training loss.

**Training Details of Linear Probe.** We freeze the pre-trained image encoder and append a linear classifier after it for linear probe classification. During training, we apply data augmentation to the input image. Concretely, we randomly crop a $224{\times}224$ patch from the input image, then conduct a random horizontal flip. During testing, we resize the shorter size to 224 then center crop a $224{\times}224$ patch as input image. We train the classifier for 90 epochs except for the ImageNet dataset, for which we train 10 epochs in total due to the large data volume. The learning rate follows a cosine decay schedule with an initial learning rate equal to 0.1. We use SGD with momentum for optimization. Weight decay is not used in our experiments. The batch size is set to 128.

## C  PROMPT ENGINEERING

Following previous works (Radford et al. (2021); Mu et al. (2022)), we extend the category names into sentences with prompts such as "a photo of {label}." before feeding them into the text encoders. For a fair comparison, we adopt the same prompts used in CLIP (Radford et al. (2021)). Specifically, for Oxford Pets, we use "a photo of a {label}, a type of pet.", while for the Food101 dataset, we use "a photo of a {label}, a type of food.". For the other datasets, we use 80 prompt templates as shown in Figure 2. For a given category name, we average the embeddings of different prompted sentences and conduct L2 normalization to obtain the final category embedding.

| | | | |
|---|---|---|---|
| a bad photo of a {label}. | a close-up photo of a {label}. | the origami {label}. | a jpeg corrupted photo of the {label}. |
| a photo of many {label}. | a black and white photo of the {label}. | the {label} in a video game. | a good photo of a {label}. |
| a sculpture of a {label}. | a painting of the {label}. | a sketch of a {label}. | a plushie {label}. |
| a photo of the hard to see {label}. | a painting of a {label}. | a doodle of the {label}. | a photo of the nice {label}. |
| a low resolution photo of the {label}. | a pixelated photo of the {label}. | a origami {label}. | a photo of the small {label}. |
| a rendering of a {label}. | a sculpture of the {label}. | a low resolution photo of a {label}. | a photo of the weird {label}. |
| graffiti of a {label}. | a bright photo of the {label}. | the toy {label}. | the cartoon {label}. |
| a bad photo of the {label}. | a cropped photo of a {label}. | a rendition of the {label}. | art of the {label}. |
| a cropped photo of the {label}. | a plastic {label}. | a photo of the clean {label}. | a drawing of the {label}. |
| a tattoo of a {label}. | a photo of the dirty {label}. | a photo of a large {label}. | a photo of the large {label}. |
| the embroidered {label}. | a jpeg corrupted photo of a {label}. | a rendition of a {label}. | a black and white photo of a {label}. |
| a photo of a hard to see {label}. | a blurry photo of the {label}. | a photo of a nice {label}. | the plushie {label}. |
| a bright photo of a {label}. | a photo of the {label}. | a photo of a weird {label}. | a dark photo of a {label}. |
| a photo of a clean {label}. | a good photo of the {label}. | a blurry photo of a {label}. | itap of a {label}. |
| a photo of a dirty {label}. | a rendering of the {label}. | a cartoon {label}. | graffiti of the {label}. |
| a dark photo of the {label}. | a {label} in a video game. | art of a {label}. | a toy {label}. |
| a drawing of a {label}. | a photo of one {label}. | a sketch of the {label}. | itap of my {label}. |
| a photo of my {label}. | a doodle of a {label}. | a embroidered {label}. | a photo of a cool {label}. |
| the plastic {label}. | a close-up photo of the {label}. | a pixelated photo of a {label}. | a photo of a small {label}. |
| a photo of the cool {label}. | a photo of a {label}. | itap of the {label}. | a tattoo of the {label}. |

Figure 2: The prompt templates used for zero-shot classification.

## D  MORE EXPERIMENTS

In this section, we further explore the impact of different hyper-parameters and design choices of DS-CLIP with more experiments.

Table 2: Different design choices of $D^3$ and SE. All models are trained on YFCC15M and evaluated on zero-shot ImageNet classification.

(a) Clustered features.

| Clustered Feature Type | Top-1 |
|---|---|
| IMAGE | **34.6** |
| TEXT | 32.8 |

(b) Ablating sample type.

| Sample Type | Top-1 |
|---|---|
| WEIGHTED | 34.3 |
| SORTED | 31.6 |
| UNIFORM | **34.6** |

(c) Ablating SE on multi-scale sample data.

| Number | SE | Top-1 | Number | SE | Top-1 |
|---|---|---|---|---|---|
| 10% | ✗ | 18.3 | 50% | ✗ | 34.6 |
|  | ✓ | **22.5** |  | ✓ | **37.9** |
| 30% | ✗ | 28.6 | 70% | ✗ | 37.6 |
|  | ✓ | **30.4** |  | ✓ | **40.2** |

(d) Ablating sample number.

| Sample Number | Top-1 |
|---|---|
| 10% | 18.3 |
| 30% | 28.6 |
| 50% | 34.6 |
| 70% | **37.6** |

(e) Ablating clustered number.

| Clustered Prototypes Number | Top-1 |
|---|---|
| 1000 | 34.2 |
| 10000 | 34.6 |
| 100000 | **34.7** |

(f) Ablating text augmentation.

| LLaVA | LLaMA | Top-1 |
|---|---|---|
| ✗ | ✗ | 34.6 |
| ✗ | ✓ | 36.5 |
| ✓ | ✗ | 37.6 |
| ✓ | ✓ | **37.9** |

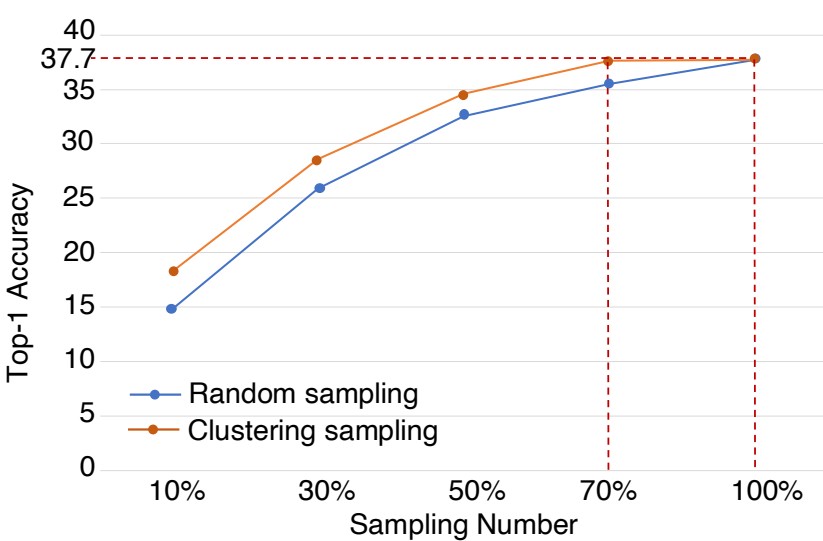

Figure 3: Zero-shot classification performance on ImageNet with different sampling strategies.

**Different clustered features and numbers in pre-training.** We cluster image-text pairs by different features (image feature and text feature) as references. Specifically, we utilize unsupervised model DINO-S/8 (Caron et al. (2021)) to extract image features for clustering, and adopt sentence transformer (Reimers & Gurevych (2019)) as the text encoder to encode text respectively. Experimental results are shown in Table 2a, CAM provides stable performance with different features for clustering, but the model with the clustering feature of the image performs better than the other.

Clustering number is another important factor, we explore three experiments of different clustering numbers. We observe a small improvement in classification accuracy as the number of clusters increased(34.2%, 34.6%, 34.7%), which had little impact as in Table 2e. We choose the number of clustering (10000) as default.

**Different sampling type and number in pre-training.** We further test different designs of sampling strategy as shown in Table 2b and 2d. For WEIGHTED, we first calculate the number of samples for each clustered category, then weighted sampling regulation by the open square proportion of the number of samples is executed (Have (2003)). For SORTED, a perspective is that image-text pairs with higher similar scores are more robust than those with lower similar scores. So we first compute the similar score of each image-text pair by extracting features from the CLIP and then choose the sample that has higher scores from the center of each cluster with corresponding proportion. We also report our default setting UNIFORM, we uniformly select the corresponding proportion of samples

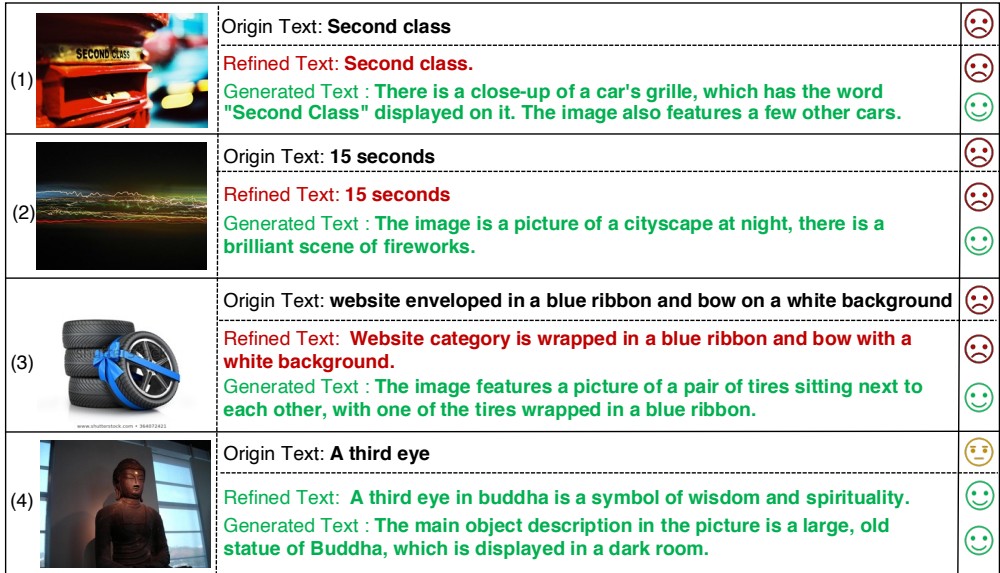

Figure 4: Samples of image-text pair with misalignment problem and generated texts from LLaMA and LLaVA.

for one epoch, and then for the next epoch, the corresponding proportion of samples are sampled from the rest in turn. We observe that all these variants produce similar results except SORTED as in Table 2b. The SORTED method selects the same samples with higher scores at each epoch, but samples of other methods are different at each epoch. Diversity of data is fully guaranteed except SORTED method.

We also test different sampling numbers during pre-training. As the number of samples increases, the performance of the model increases as in Table 2d.

**Clustering with different sample ratios.** Moreover, by utilizing different sampling ratios, the model trained according to the clustered sampling strategy outperforms the model of random sampling as shown in Figure 3. We can attribute this better performance to the diversity of samples maintained in the training epoch with the clustered dataset.

## E    MORE VISUALIZATION

In this section, we provide more visualization results of augmented texts. As shown in Figure 4, most of the origin samples are inconsistent between image and text. The origin text of sample (1) is the "word" on the image, the origin texts of sample (2) and sample (3) do not describe the main content of the image. The origin text of sample (4) describes the image as incomplete. The refined text based on LLaMA from samples (1) to (3) is inaccurate but the refined text in the sample (4) is a complete supplement. However, all generated texts from LLaVA can accurately describe the content of the image.

Although the augmented text from Text2Text rewriter sometimes may not return texts that provide descriptions about the related-image contents, the proposed DS-CLIP is also robust to feature representation learning. Since the proposed DS-CLIP not only depends on the origin text and augmented texts from LLaMA but also takes generated texts from LLaVA into consideration, it is more robust than origin CLIP (Radford et al. (2021)) trained with only one image-text pair.

## F    DISTRIBUTOIN OF VISUAL EMBEDDING SPACE BY T-SNE

To gain a deeper understanding of the distinctions between the features learned from DS-CLIP and vanilla CLIP, we provide distribution of visual embedding space with CLIP and DS-CLIP on 10 classification datasets using t-SNE (Maaten & Hinton (2008)) in Figure 5. We visualize the

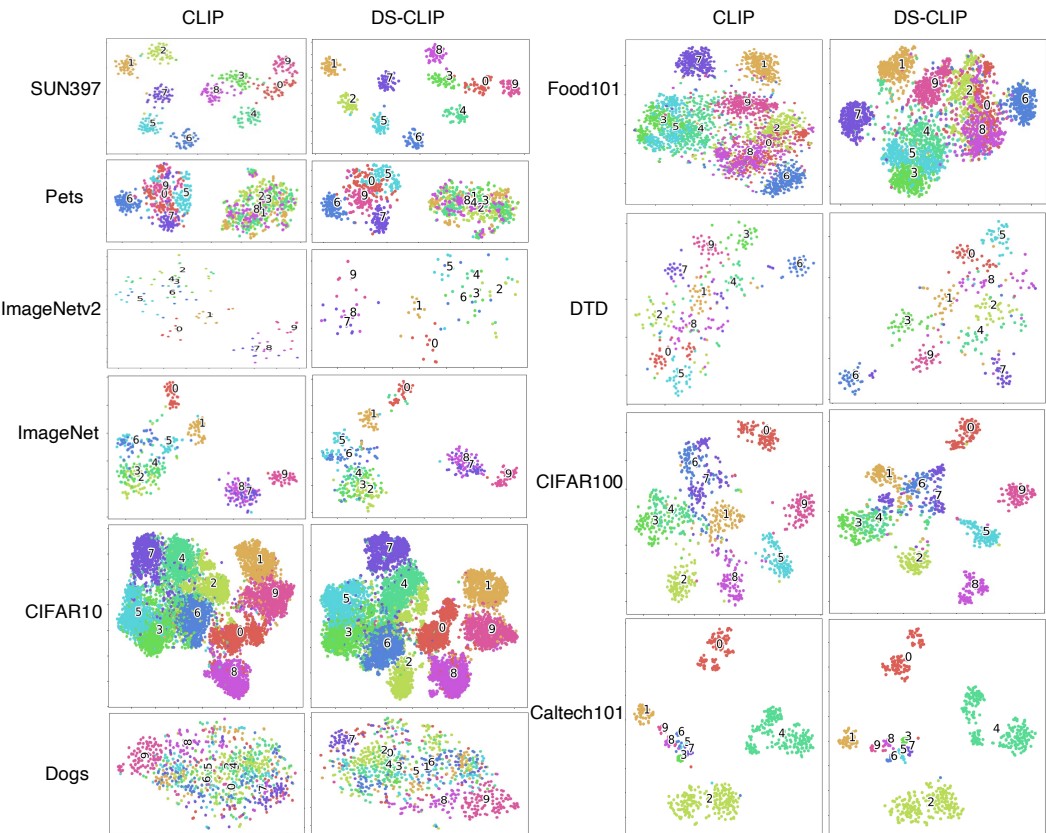

Figure 5: Distribution of visual embedding space with CLIP and DS-CLIP on 10 classification datasets using t-SNE.

features from the first 10 classes for each dataset. From Figure 5, we observe that DS-CLIP trained with augmented texts has more distinguished boundaries and is more compact for each class. This observation suggests that augmented text helps learning an effective image embedding space which is also well-suited for downstream tasks.