# OpenReview forum: "Data De-Duplication and Semantic Enhancement for Contrastive Language-Image Pre-training"
_ICLR.cc/2024/Conference — Submitted to ICLR 2024_

### Official Review · Reviewer_xVnC · 2023-10-17

**Soundness:** 2 fair
**Presentation:** 2 fair
**Contribution:** 2 fair
**Rating:** 5
**Confidence:** 5

**Summary:**

This paper explores several tricks to enhance the CLIP model.
These tricks include cluster-based data de-duplication, text augmentation with LLM and VLM, and image augmentation.
The pre-training is performed on the large-scale Laion400M dataset.
With the experimental results on a wide variety of downstream tasks, we can observe that the proposed method achieves improved performance over the plain CLIP model.

**Strengths:**

- The paper is well-written and well-organized.
Most parts of this paper are easy to follow and understand.

- The proposed method achieves consistent improvements on diverse downstream tasks and datasets when compared to the vanilla CLIP model.

- The authors conducted detailed ablation studies of the proposed method.

**Weaknesses:**

- The biggest concern for this paper is the key intuition and motivation of the proposed method.
The data de-duplication is leveraged to reduce the training samples, which is useful for training efficiency.
However, the other three tricks mostly focus on augmentation, thus introducing more data for training.
This mix-up makes the readers follow the key contribution of this paper.

- The novelty of the proposed method is somewhat limited.
All these approaches look like tricks that have been well-explored by existing literature.

- The first approach, i.e., data de-duplication is also limited by the pre-trained vision model.
If we use another model rather than DINO, the sampled images could be different, which may lead to different conclusions.

- There are many notational errors (for example, N or B for the number of pairs?) in the descriptions of Sec. 3.1. Please carefully revise them.

**Questions:**

- Eqn.1 seems not right.
Normally the NCE contrastive loss only holds one positive label.
But for this approach, there could be at least three positive labels.
Maybe some theoretical analysis helps address this concern.

- Have the authors also considered generating images as augmentation?

---

> ### Author Response · Authors · 2023-11-18
> **Response to xVnC**
>
> Thank you for your insightful advice and valuable questions, we will respond to your concerns point by point.
>
> > **Q1: The key intuition and motivation of the proposed method. Explanation of the contradiction between the effectiveness of ($\text{D}^3$) and the other three modules e.g. text augmentation, DCTM, and MSTM.**
>
> The key intuition and motivation of the proposed method is that construct a semantically diverse multimodal dataset and present an efficient training algorithm for better representation learning.
>
> The generated texts are completed by one-time processing, which can be reused to train other models, the time does not require double-counting. The efficient training algorithm is proposed to utilize semantically enriched datasets. We also provide the specific calculation time for each module.
>
> Specifically, as Q4 of review Fx4U, we counted the time for all modules. It takes about 175 hours to rewrite one text from LLaMA for the entire Laion400M dataset on 32 A100 GPU machines, and it takes only 75 hours to generate texts from LLaVA with batch inference. Training baseline CLIP on LAION-400M with large ViT-L/14 almost takes 550 hours on 32 A100 GPU machine. Training with half of the clustered augmented data can achieve comparable performance but its training time is reduced by half. Clustering time is negligible as compared to text generating, it only takes a few hours on one A100 GPU machine. The total training cost has decreased a little overall (550 vs 525) on 32 A100 GPU machines.
>
>
> > **Q2: The limination of novelty.**
>
>
> Please refer to General Response.
>
> > **Q3: The effect of the pre-trained model e.g. DINO in data de-duplication.**
>
> ###### 						Tab.7: Ablation study for clustering pre-trained model on ImageNet(IN) datasets for Zero-shot image classification (%), 																							all experiments are used 50% clustered data.
>
> | Evaluation Type | Clustering Pre-trained Model | IN |
> | :-: | :-: | :-: |
> | Zero-shot | Dino | 34.6 |
> | Zero-shot | MAE | 34.2 |
>
>
> As per your suggestion, we explored the use of different models to extract image features for clustering e.g. MAE and Dino.
>
> As shown in the above table, there is no big gap in classification performance between using pre-trained model MAE to extract image features and using Dino to extract image features for clustering. As long as with a well-pre-trained model, the image feature for clustering has little impact, all of which can produce better clustering results.
>
> > **Q4: Corrected notational errors.**
>
>
> Thanks for your kind suggestion, we have modified the paper in detail.
>
> > **Q5: Normally the NCE contrastive loss only holds one positive label. But for this approach, there could be at least three positive labels. Analysis of Eqn.1.**
>
>
> The NCE contrastive loss only holds one positive label. According to SE in DS-CLIP, one image has $M+1$ multiple generated texts represented as in Eqn.1, which constructs $M+1$ image-text pairs. In Eqn.1, for each image corresponding to $M+1$ multiple texts, we calculate the mean of contrastive loss of $M+1$ image-text pairs during training. In all, Eqn.1 can be interpreted as the mean of multiple NCE contrastive losses.
>
> > **Q6: Generating images as augmentation is used for DS-CLIP.**
>
>
> The pre-trained dataset for DS-CLIP is YFCC15M, whose text description is too short and unable to adequately describe the image content. The image generated according to the description of the short text is quite different from the original image, which leads to the inconsistency of the sample being self-supervised during the training. The generated images are shown in the following link. [generate_img](https://clip-multimodal.oss-cn-beijing.aliyuncs.com/siyang/paper/ICLR2024/generated_image.png?OSSAccessKeyId=LTAI5tGRJ62Fo4Ly48B8Zd1g&Expires=5300205749&Signature=udAeVnUz2UwM8MQBhNmkudbTRXU%3D)
>
> Training with the generated images in DS-CLIP instead of the augmented image through a transformer, and other modules and parameters are the same with Experiment ID 6 in Table 1 in the paper, the performance decline is more serious(from 39.3% to 30.1%).

---

### Official Review · Reviewer_Fx4U · 2023-10-25

**Soundness:** 2 fair
**Presentation:** 3 good
**Contribution:** 2 fair
**Rating:** 5
**Confidence:** 4

**Summary:**

This paper proposes a novel training strategy called DS-CLIP to improve the traditional contrastive language-image pre-training (CLIP) model. It introduces two components - Data De-Duplication (D3) and Semantic Enhancement (SE) to reduce training costs and enhance dataset diversity. D3 employs data clustering and sampling to reduce scene redundancy without losing diversity. SE uses large language models to generate diverse, semantically enriched captions to address image-text misalignment. Furthermore, this paper proposes Diverse Captions Training and Modality Self-enhancement Training for effective learning. Extensive experiments show DS-CLIP achieves state-of-the-art on various downstream tasks, including classification, retrieval, detection and segmentation.

**Strengths:**

(1)	This paper addresses the efficiency problem of CLIP pre-training by proposing Data De-Duplication (D3) and Semantic Enhancement strategies, which first clusters and re-samples noisy multi-modal data to ensure a balanced semantic distribution without reducing the scene diversity, then employ powerful LLM and VLLM to enrich semantic information of text and mitigate the issue of text-image misalignment.

(2)	This paper presents a one-to-multiple mapping among image and text as the Diverse Captions Training Mechanism (DCTM) and Modality Self-enhancement Training Mechanism (MSTM), which effectively reduces training time and alleviates data redundancy and misalignment.

(3)	This paper is clearly written and easy to follow. The problems and limitations of previous CLIP training are clearly explained. The method section has explained technical details well. The related tables and figures also are presented clearly.

(4)	Extensive experiments have shown that the DS-CLIP significantly outperforms traditional CLIP on various vision-language tasks, especially fine-grained classification datasets, and various patch-level downstream tasks from 0.2% to 23.9%, with ONLY half of the training time.

**Weaknesses:**

(1)	The core contributions of this paper are the D3 and SE modules, which belong to the data augmentation and data cleaning essentially. The clustering, re-sampling, and text re-generation are all very common strategies in recent work, e.g., BLIP [1], and BLIP-2[2]. Hence the technical contribution is weak.

(2)	The previous contrastive loss function can deal with the multi-positive image-text pairs during training. However the experiments lack ablation studies or theoretical justification, more analysis can help prove the effectiveness of the proposed loss function.

(3)	The CLIP is a famous multi-modal pretraining model. However, this paper only contains pure vision-understanding tasks and lacks sufficient experiments on various multi-modal tasks and datasets, e.g., image-text matching, video-text retrieval, and image captioning.

[1] Bootstrapping Language-Image Pre-training for Unified Vision-Language Understanding and Generation, ICML, 2022

[2] BLIP-2: Bootstrapping Language-Image Pre-training with Frozen Image Encoders and Large Language Models, 2023

**Questions:**

(1)	This paper has claimed that the DS-CLIP only needs half the training time compared with traditional CLIP. But the D3 and SE modules also need large computational costs, e.g., the clustering operation, and the inference process of LLM/VLLM. It is necessary to report related time costs since the extra data augment is an important part of the proposed method.

(2)	The Data De-Duplication (D3) relies on the clustering algorithm to converge unlabeled data. However, the K-means is not a good choice for large amounts of data, in which the runtime and memory cost are non-negligible with multiple iterations. Have you tried any other clustering algorithms, e.g., spectral clustering? Besides, why not cluster the texts?

---

> ### Author Response · Authors · 2023-11-18
> **Response to Fx4U (Part 1)**
>
> Thank you for your insightful advice and valuable questions, we will respond to your concerns point by point.
>
> > **Q1: The clustering, re-sampling, and text re-generation are all very common strategies in recent work,e.g., BLIP [1], and BLIP-2[2]. Hence the technical contribution is weak.**
>
> **BLIP[1]** proposes an efficient utilization method of web noisy data. It is first trained with noisy data, and then BLIP is used to generate a series of subtitles through a pre-trained Captioner, moreover, the generated subtitles are filtered through a pre-trained Filter, deleting the noisy subtitles from the original web and synthetic texts to get clean data. Finally, BLIP is trained again with clean data. The re-sampling in BLIP is filtered and text re-generation is iteratively updated as model training.
>
> **In contrast to the BLIP**, our method uses the rich knowledge of LLM/VLLM to enhance the text data and rewrite the data semantically and structurally without repeated model training. We pay more attention to constructing a more diverse and semantic dataset from the web noisy dataset rather than the training strategy. The re-sampling is without filtering data and keeps the diversity of the dataset at each training epoch. Additionally, we conducted experiments comparing the training approaches of initializing the model with the backbone of the VLLM itself and training the multimodal representation model using the data generated by the LLM and VLLM. We find that utilizing the augmented data from a well-trained LLM and VLLM yields superior performance, further confirming the importance of LLM and VLLM in data construction. We hope our augmented dataset will help further research and pay more attention to the data construction by LLM and VLLM.
>
> **BLIP-2[2]** proposes an efficient pre-trained algorithm through training Q-Former to align a visual model and a Large Language Model (LLM) of freezing parameters. The Q-Former connects an image encoder of a visual model and an LLM encoder. In BLIP-2, the data is not extended or augmented from source, and the training parameters are reduced from the model perspective for efficient training.
>
> **The essential difference with our method** is that we start from the misaligned between image and text, and use the rich knowledge of LLM/VLLM to enrich text representation. We focus on the problem of data noise from the source rather than the training strategy. We utilize Data De-Duplication ($\text{D}^3$) based on augmented diverse data to achieve superior performance with fewer computing resources. $\text{D}^3$ is different from data cleaning, we do not filter data during training but sample a certain percentage of data points from each pre-clustered center. The text re-generated and sampling strategy are quite different from BLIP[1] and BLIP-2[2]. Extensive experiments show that the constructed dataset is helpful in improving the performance of the current modal e.g. DeCLIP, LaCLIP. Please refer to the question 6 of review YX2M.
>
> **Reference:**
>
> [1] Li J, Li D, Xiong C, et al. Blip: Bootstrapping language-image pre-training for unified vision-language understanding and generation[C]//International Conference on Machine Learning. PMLR, 2022: 12888-12900.
>
> [2] Li J, Li D, Savarese S, et al. Blip-2: Bootstrapping language-image pre-training with frozen image encoders and large language models[J]. arXiv preprint arXiv:2301.12597, 2023.
>
> > **Q2: The previous contrastive loss function can deal with the multi-positive image-text pairs during training. More analysis to verify the effectiveness of the proposed loss function.**
>
> The previous contrastive loss function deals with the multi-positive image-text pairs during training as follows.
>
> **In BLIP[1]**, each batch ensures only one positive sample during training.
>
> **In ALBEF[3]**, if there are K-positive samples in batch, then the gt at the corresponding position of contrastive loss is $1/K$.
>
> **In our DS-CLIP**, for each image corresponding to $M+1$ multiple texts, we calculate the mean of contrastive loss of all $M+1$ image-text pairs.
>
> In our ablation study as shown in Experiment ID 4 in Table 1, only one text is sampled from multiple generating texts with the standard text-image contrastive loss as same as BLIP[1].  Experiment ID 5 in Table 1 shows that multiple text embeddings for DCTM bring a +0.6% improvement in the accuracy of classification. We add an experiment using the contrastive loss in ALBEF[3] instead of DCTM, other parameters are kept the same. The performance with contrastive loss in ALBEF[3] is 38.0% on image classification, which has a comparable result with BLIP[1] as Experiment ID 4 in Table 1. This analysis is presented in Sec4.2.
>
> **Reference:**
>
> [3] Li J, Selvaraju R, Gotmare A, et al. Align before fuse: Vision and language representation learning with momentum distillation[J]. Advances in neural information processing systems, 2021, 34: 9694-9705.

---

> > ### Author Response · Authors · 2023-11-18
> > **Response to Fx4U (Part 2)**
> >
> > > **Q3: More experiments on image-text matching, video-text retrieval, and image captioning.**
> >
> > Thanks for your suggestion. We added more experiments on image-text matching, video-text retrieval, and image captioning.
> >
> > **For the image-text matching task**, we conducted image-text retrieval on COCO and Flickr30K datasets, as shown in Sec.4.3 in the paper.
> >
> > In order to verify that the image-language model (DS-CLIP) has a strong generalization ability for video-language tasks, we test DS-CLIP on a 1k test split of MSR-VTT for **video-text retrieval tasks**. The evaluation indicator of R@1 is the same as image-text retrieval. For video-text retrieval, zero-shot DS-CLIP outperforms baseline CLIP as shown in the following table.
> >
> > **For the image captioning task**, we consider the COCO dataset with BLEU@4 and CIDEr indicator. We use image encoder VIT-B/16 of CLIP and DS-CLIP to initialize the image encoder of coca[1] for fair comparison following the original code training on the COCO caption trainset. The result is shown in the following table, we can find out DS-CLIP far exceeds baseline CLIP by 1.7% of BLEU@4 and 4.2% of CIDEr.
> >
> > ###### 		Tab.5: Performance of DS-CLIP on downstream tasks (Video-to-Text(V2T) Retrieval, Image Caption, and Image-to-Text(I2T) Retrieval) (%).
> >
> > | Method | V2T Retrieval |  | Image Caption |  | I2T Retrieval |  |  |  |
> > | --- | --- | --- | --- | --- | --- | --- | --- | --- |
> > |  | MSR-VTT |  | COCO |  | COCO |  | Flickr30K |  |
> > |  | V2T/R@1 | T2V/R@1 | BLEU@4 | CIDEr | I2T/R@1 | T2I/R@1 | I2T/R@1 | T2I/R@1 |
> > | CLIP | 9.1 | 9.1 | 33.7 | 108.3 | 20.4 | 33.3 | 37.8 | 57.3 |
> > | DS-CLIP | 15.1 (+6.0) | 15.1 (+6.0) | 35.4 (+1.7) | 112.5 (+4.2) | 30.7 (+10.3) | 45.5 (+12.2) | 52.3 (+14.5) | 75.3 (+18.0) |
> >
> >
> > **Reference:**
> >
> > [1] Yu J, Wang Z, Vasudevan V, et al. Coca: Contrastive captioners are image-text foundation models[J]. arXiv preprint arXiv:2205.01917, 2022.
> >
> > > **Q4: Report of related time costs of each extra data augment module.**
> >
> > As shown in Sec4.2, **''DS-CLIP is scalable for training on large datasets and models''**, we report the inference time of SE.
> >
> > It takes about 175 hours to rewrite one text from LLaMA for the entire Laion400M dataset on 32 A100 GPU machines, and it takes only 75 hours to generate texts from LLaVA with batch inference on 32 A100 GPU machines. Training baseline CLIP on LAION-400M with large ViT-L/14 takes 550 hours on 32 A100 GPU machines. Training with half of clustered augmented data can achieve comparable performance but its training time is reduced by half. Clustering time is negligible as compared to text generating, it only takes a few hours on one A100 GPU machine. The total training cost has decreased a little overall(550 vs 525) on 32 A100 GPU machines.
> >
> > In addition, the generated texts are completed by one-time processing, which can be reused to train other models, the time does not require double-counting.
> >
> > > **Q5: Other clustering algorithms experiments, e.g., spectral clustering in our experiments.**
> >
> > Thanks for your advice. We use spectral clustering instead of K-means to pre-process the YFCC15M dataset, keeping other parameters are same. The clustering type has little impact on the performance of DS-CLIP as shown in the following table, the results are almost the same(34.6% vs 34.5%).
> >
> > ###### Tab.6: Ablation study for Clustering type and feature on ImageNet(IN) datasets for Zero-shot image classification (%), all experiments are used 																													50% clustered data.
> >
> > | Evaluation Type | Clustering Feature | Clustering Type | IN |
> > | :-: | :-: | :-: | :-: |
> > | Zero-shot | Image | K-means | 34.6 |
> > | Zero-shot | Image | Spectral | 34.5 |
> > | Zero-shot | Text | K-means | 32.8 |
> >
> >
> > > **Q6: Using the clustering text in our experiments.**
> >
> >
> > We report the results of using the text clustering in Appendix.D, the model with the clustering feature of the image performs better than the text clustering(34.6% vs 32.8%), which is also shown in the above table.

---

> ### Comment · Reviewer_Fx4U · 2023-11-21
> **Comments by Reviewer Fx4U**
>
> Thanks to the author for the reply. I carefully read the author's response and the comments of other reviewers. My main concern, similar to reviewer AY1m's, was not well addressed. Current explanations about novelty are still weak. So I changed my rating to 5.

---

> > ### Author Response · Authors · 2023-11-21
> > **Response to Fx4U**
> >
> > Thanks for your valuable and constructive question.
> >
> > As answered to the Reviewer AY1m, a significant contribution of our work, which is absent in BLIP, is as follows:
> > We validate that models trained on LLaVA synthetic data and original data can recognize more categories than LLaVA itself.
> > Specifically, since synthetic data relies on an MLLM (e.g., LLaVA), it is limited to describing only the categories that the MLLM is aware of.
> > Hence, a crucial premise for using synthetic data is to demonstrate that introducing synthetic data helps the model recognize more categories than the pre-trained MLLM.
> > Our paper provides empirical evidence to support this claim through experiments.
> > As shown in Table 1, our best model achieves an impressive accuracy of 78.7\% on ImageNet zero-shot classification, surpassing the performance of LLaVA-initialized CLIP methods (76.0\%) by a significant margin.
> >
> > In contrast, BLIP does not address this aspect in their paper.
> > They solely focus on cross-modal retrieval and captioning on the COCO and Flickr datasets.
> > However, both COCO and Flickr contain a significantly smaller number of categories compared to ImageNet.
> > Thus, their experiments do not address the crucial question of whether synthetic data can improve model performance on large vocabulary recognition tasks, such as ImageNet zero-shot classification.

---

### Official Review · Reviewer_AY1m · 2023-10-31

**Soundness:** 2 fair
**Presentation:** 3 good
**Contribution:** 2 fair
**Rating:** 5
**Confidence:** 4

**Summary:**

This paper introduces a new training framework for CLIP-like models, aiming to 1) reduce training costs and 2) mitigate the misalignment issues stemming from noisy image-text pairs. For this, the authors propose following components:

1. Data De-duplication (D3) enables fast training without losing the diversity of sampling by leveraging pre-clustered prototypes which enables.

2. Semantic Enhancement (SE) mitigates the noisy image-text correspondence issues by generating more descriptive captions with powerful pre-trained Large Language Models (LLMs) and Vision-Language Large Models (VLLMs)

3. Diverse Captions Training Mechanism (DCTM) and a Modality Self-enhancement Training Mechanism (MSTM) : DCTM utilizes diverse captions, while MSTM employs a combination of uni-modal contrastive learning.

 As a result, it achieves state-of-the-art performance over various downstream tasks with half of the training time compared with original CLIP.

**Strengths:**

1. The paper is well-written and figures are easy to understand.
2. The motivation of paper (efficient pre-training by mitigating mis-alignment in image-text papers and scene redundancy) is solid.
3. The experimental results are strong.

**Weaknesses:**

Despite  strong experimental results and motivation, the novelty of the proposed methods appears to be limited:

   1) In SE: The effectiveness of synthetic captions from VLP models for mitigating noisy image-text alignment has already been demonstrated by BLIP.  Therefore, it is somewhat straightforward that more descriptive captions from recent LLaVA models would be effective. Furthermore, as the authors themselves pointed out, the concept of using LLM-generated captions has already been proposed in LaCLIP. Moreover, the effectiveness of using both LLaVA and LLaMA is unclear. See question 2.

   2) In DCTM: Previous works like OSCAR [1], ALBEF [2], and BLIP have empirically shown that diverse captions (one image with multiple captions) from sources like COCO and Flickr are effective in enhancing performance. These works treat each image-caption pair as unique; for instance, if one image comes with five captions as in the COCO setting, they construct five distinct pairs. The difference in the current approach is the use of diverse captions with a multi-positive contrastive loss. However, it remains unclear where the benefits of this approach specifically originate from. See question 1.

   3) In MSTM: The utility of uni-modal contrastive losses in improving performance has already been showcased by ERNIE-VIL 2.0 [3].


[1] Li, Xiujun, et al. "Oscar: Object-semantics aligned pre-training for vision-language tasks." Computer Vision–ECCV 2020: 16th European Conference, Glasgow, UK, August 23–28, 2020, Proceedings, Part XXX 16. Springer International Publishing, 2020.

[2] Li, Junnan, et al. "Align before fuse: Vision and language representation learning with momentum distillation." Advances in neural information processing systems 34 (2021): 9694-9705.

[3] Shan, Bin, et al. "ERNIE-ViL 2.0: Multi-view Contrastive Learning for Image-Text Pre-training." arXiv preprint arXiv:2209.15270 (2022)

**Questions:**

1)  The benefits from DCTM comes from the data augmentations (use multiple captions) or from multi-positive contrastive loss? Moreover, what is the difference between multi-positive contrastive loss and supervised contrastive loss [4]? What is the advantage of using multi-positive contrastive loss?

2)  Does it have to use both LLaVA and LLaMA? In table 2 (c), the gap between LLaVA only and LLaVA/LLaMA seems very marginal. Isn't it possible to use LLaVA only to generate diverse captions with proper prompts?

3) In Figure 3 and Figure5, it seems that the boundaries are still indistinguishable.

[4] Khosla, Prannay, et al. "Supervised contrastive learning." Advances in neural information processing systems 33 (2020): 18661-18673.

---

> ### Author Response · Authors · 2023-11-18
> **Response to  AY1m (Part 1)**
>
> Thank you for your insightful advice and valuable questions, we will respond to your concerns point by point.
>
> > **Q1: Limitation of the novelty of the proposed methods. Different contributions of our method compared with BLIP and LaCLIP.**
>
> **BLIP[1]** proposes an efficient utilization method of noisy web data. It is first trained with noisy data, and then BLIP is used to generate a series of subtitles through a pre-trained Captioner, next, the generated subtitles are filtered through a pre-trained Filter, deleting the noisy subtitles from the original web and synthetic texts to get clean data. Finally, BLIP is trained again with clean data.
>
> **In contrast to the BLIP**, our method uses the rich knowledge of LLM/VLLM to enhance the text data and semantically and structurally rewrite the data without repeated model training. We pay more attention to constructing a more diverse and semantic dataset from the web noise dataset rather than the training strategy. Additionally, we conducted experiments comparing the training approaches of initializing the model with the backbone of the VLLM itself and training the multimodal representation model using the data generated by the LLM and VLLM. We find that utilizing the augmented data from a well-trained LLM and VLLM yields superior performance, further confirming the importance of LLM and VLLM in data construction.
>
> **LaCLIP[2]** rewrites and changes the structure of text from the original text using LLaMA and chatbots but maintains semantic information of the text. Text rewritten in LaCLIP only ensures the semantics associated with the corresponding original text but it does not address the misalignment problem of image and text.
>
> **We can supplement the shortcomings of LaCLIP** from the aspects of text augmented via VLLM. We combine our generated data with LaCLIP, and the average mAP on 10 image classification datasets is improved by 0.6% with our augmented data. Please refer to question 6 in the review YX2M.
>
> **Reference:**
>
> [1] Li J, Li D, Xiong C, et al. Blip: Bootstrapping language-image pre-training for unified vision-language understanding and generation[C]//International Conference on Machine Learning. PMLR, 2022: 12888-12900.
>
> [2] Fan L, Krishnan D, Isola P, et al. Improving CLIP Training with Language Rewrites[J]. arXiv preprint arXiv:2305.20088, 2023.
>
> > **Q2: Explanation of the effectiveness of using both LLaVA and LLaMA. Verify that it is possible to use LLaVA only to generate diverse captions with proper prompts.**
>
> Thanks for your suggestion. The pre-trained dataset for DS-CLIP is YFCC15M in our experiments, whose text description is too short, and the generated texts according to LLaMA have fewer variations. Using LLaMA to generate text has less improvement on YFCC15M.
>
> 1. We also pre-train DS-CLIP based on ViTB/16 on CC3M with SE as shown in the following table. The text description of CC3M is longer and more abundant. The improvement is respectively 4.7% and 5.8% based on LLaMA and LLaVA. On the basis of LLaVA, the performance continues to increase by 1.6% combined with LLaMA.
>
> 2. We design several proper prompts for LLaVA to generate texts such as ''Can you explain this image in detail?'', ''Tell me in detail what is main object description in this picture.'' and ''What happened in the picture?'' Using these proper prompts to generate three texts for each image with LLaVA on the CC3M dataset, we conduct an experiment named LLaVA+ in the following table. The performance improves by 0.4% from 21.6% to 22.0% compared with text augmented from LLaVA, but it is still lower than text augmentation from LLaVA and LLaMA. The generated text from LLaVA has already described the image content in as much detail as possible, with more prompts to generate texts has fewer improvements.
>
> ###### 								Tab.3: Ablation study with text augmentation on ImageNet(IN) for zero-shot image classification (%)
>
> | LLaMA | LLaVA | LLaVA+ | Top-1 |
> | :-: | :-: | :-: | :-: |
> | ✗ | ✗ | ✗ | 15.8 |
> | ✓ | ✗ | ✗ | 20.5 |
> | ✗ | ✓ | ✗ | 21.6 |
> | ✓ | ✓ | ✗ | 23.2 |
> | ✗ | ✗ | ✓ | 22.0 |

---

> > ### Author Response · Authors · 2023-11-18
> > **Response to AY1m (Part 2)**
> >
> > > **Q3: The benefits from DCTM come from the data augmentations (use multiple captions) or from multi-positive contrastive loss.**
> >
> > In our experiment, the data augmentations (use multiple captions) and multi-positive contrastive loss are all helpful in improving DS-CLIP performance.
> >
> > As shown in Tab.1, Experiment ID 4 represents that one text is sampled from multiple generating captions with the standard text-image contrastive loss. The performance of DS-CLIP is improved by 3.3% compared with without augmented captions as Experiment ID 3. Furthermore, we utilize multi-positive contrastive loss in DCTM for training as Eqn.1, Experiment ID 5 in Tab. 1 shows that multi-positive contrastive loss used in DCTM brings +0.6% improvement in the accuracy of classification. Therefore, using multiple captions and multi-positive contrastive loss in DCTM could implicitly help the model to improve the data diversity and align the image and text embeddings more efficaciously. These analyses are represented in Sec4.1 in the paper.
> >
> > > **Q4: The difference between multi-positive contrastive loss and supervised contrastive loss, and the advantage of using multi-positive contrastive loss.**
> >
> >
> > - **Supervised contrastive loss**[3] introduces a **fully supervised** setting into contrastive loss. It pulled samples with the same class together in embedding space and pushed apart clusters of samples from different classes combined with contrastive loss.
> >
> > - **Multi-positive contrastive loss** leverages the additional and richer supervision from the language modality to align the formation of image-text embeddings, which allows each image to be paired with all of the diverse texts describing its content. Training with multi-positive contrastive loss in our method is without requiring the label data. The main consideration is that of the alignment between image and text in cross-mode, in addition, we utilize MSTM to significantly enhance uni-modal representation.
> >
> > Through data clustering operation in DS-CLIP, each sample has a clustered label, and the supervised contrastive loss can also be used in DS-CLIP, the performance of DS-CLIP with combining supervised contrastive loss is further improved by 0.5% as shown in the following table. Multi-positive contrastive loss and Supervised Contrastive loss can be used together.
> >
> > The advantage of multi-positive contrastive loss could implicitly help the model to align the image and text embeddings more efficaciously with diverse potential information of data. In future work, we will consider supervised contrastive loss in our method.
> >
> > ###### 								Tab.4: Ablation study with different contrastive losses on ImageNet(IN) for zero-shot image classification (%)
> >
> > | DSTM | supervised contrastive loss | Top-1 |
> > | :-: | :-: | :-: |
> > | ✓ | ✗ | 38.5 |
> > | ✗ | ✓ | 38.2 |
> > | ✓ | ✓ | 39.0 |
> >
> > **Reference:**
> >
> > [3] Khosla P, Teterwak P, Wang C, et al. Supervised contrastive learning[J]. Advances in neural information processing systems, 2020, 33: 18661-18673.
> >
> > > **Q5: Limitation of MSTM: Uni-modal contrastive loss is showcased by ERNIE-VIL 2.0 [4].**
> >
> > **ERNIE-VIL 2.0** proposed multi-view learning with image-to-text, text-to-image, image-to-image, and text-to-text contrastive loss, which is single and one-to-one self-supervision loss in-mode.
> >
> > **Different from ERNIE-VIL 2.0**, MSTM is a multi-positive contrastive loss. We naturally utilize multi-positive contrastive loss to the pre-train model because of the generated multiple captions.
> >
> > In particular, one image has $M+1$ multiple texts and each image with one corresponding text calculates a standard self-supervision loss, then the mean of $M+1$ Multiple Self-supervision Losses is the final loss
> >
> > The purpose of MSTM is to enhance uni-modal representation significantly, and the performance of DS-CLIP with MSTM is greatly promoted as Experiment ID 6 in Tab.1 in the paper.
> >
> > **Reference:**
> >
> > [4] Shan B, Yin W, Sun Y, et al. ERNIE-ViL 2.0: Multi-view Contrastive Learning for Image-Text Pre-training[J]. arXiv preprint arXiv:2209.15270, 2022.
> >
> > > **Q6: Boundaries are still indistinguishable in Figure 3 and Figure 5.**
> >
> > The performance of DS-CLIP has significantly improved compared with baseline CLIP on various tasks. t-SNE is utilized to measure the quality of features learned. Figure 3 and Figure 5 show all classification datasets, we have a significant improvement and clear boundary for class 3 and class 5 in Figure 3.
> >
> > However, such as ImageNetv2, pets, dogs, and DTD, which have a limited number of samples per class in the test set, making it difficult to generate more obvious visualizations. Other datasets contain enough samples per class (e.g. SUN397, Food101, CIFAR10, CIFAR100), which still have better boundaries and more distinct clusters compared to baseline CLIP. The results on the evaluation indicators are more intuitive to verify the effectiveness of DS-CLIP.

---

> ### Comment · Reviewer_AY1m · 2023-11-21
> **Response to author**
>
> Thanks for detailed explanation.
>
> Comparison with BLIP
>
>
> --> In my review, my main point was that the concept of leveraging synthetically generated captions is the same as BLIP.
>       In their setup, since they did the first work to use generated captions, it is natural to require a training phase for captioner.
>       It is not author's contribution which do not require repetition step since authors simply assume that pre-trained model like LLaVA is available.
>       Moreover, it is really straightforward leveraginng captions from more recent LLaVA which can generate more descriptive captions (which leverages the power of LLM) is much more effective than captions from BLIP. Since the performance increase is mainly comes from SE, my main concern about novelty and contribution is not resolved yet.

---

> > ### Author Response · Authors · 2023-11-21
> > **Response to AY1m**
> >
> > Thanks for your valuable and constructive question.
> >
> > A significant contribution of our work, which is absent in BLIP, is as follows:
> > We validate that models trained on LLaVA synthetic data and original data can recognize more categories than LLaVA itself.
> > Specifically, since synthetic data relies on an MLLM (e.g., LLaVA), it is limited to describing only the categories that the MLLM is aware of.
> > Hence, a crucial premise for using synthetic data is to demonstrate that introducing synthetic data helps the model recognize more categories than the pre-trained MLLM.
> > Our paper provides empirical evidence to support this claim through experiments.
> > As shown in Table 1, our best model achieves an impressive accuracy of 78.7\% on ImageNet zero-shot classification, surpassing the performance of LLaVA-initialized CLIP methods (76.0\%) by a significant margin.
> >
> > In contrast, BLIP does not address this aspect in their paper.
> > They solely focus on cross-modal retrieval and captioning on the COCO and Flickr datasets.
> > However, both COCO and Flickr contain a significantly smaller number of categories compared to ImageNet.
> > Thus, their experiments do not address the crucial question of whether synthetic data can improve model performance on large vocabulary recognition tasks, such as ImageNet zero-shot classification.

---

> > > ### Comment · Reviewer_AY1m · 2023-11-21
> > > **Response to author**
> > >
> > > Thanks for the explanation,
> > >
> > > I'm not sure that "our best model achieves an impressive accuracy of 78.7% on ImageNet zero-shot classification, surpassing the performance of LLaVA-initialized CLIP methods (76.0%) by a significant margin." is a fair setting.
> > >
> > > First of all, isn't it that LLaVA freezes CLIP in training? Thus, the CLIP from LLaVA is almost the same as the original CLIP except for the linear layer.
> > >
> > > Secondly, what does it mean that COCO and Flickr have a significantly smaller number of categories? Do you mean cluster? Since they do not have class information, each instance corresponds to a distinct class.
> > >
> > > Thirdly, as the authors pointed out, the experimental results from BLIP are absent in your experiment. We cannot be sure whether their approach is effective or not.
> > >
> > > Finally, I'm not sure that "whether synthetic data can improve model performance on large vocabulary recognition tasks, such as ImageNet zero-shot classification" is a critical question. It seems a natural extended task of retrieval and captioning tasks.

---

> > > > ### Author Response · Authors · 2023-11-22
> > > > **Response to AY1m**
> > > >
> > > > Thank you for your further insightful advice and valuable questions.
> > > >
> > > > > I'm not sure that "our best model..." is a fair setting. First of all, isn't it that LLaVA freezes CLIP in training? Thus, the CLIP from LLaVA is almost the same as the original CLIP except for the linear layer.
> > > >
> > > >
> > > > The purpose of our experiment is to verify that models trained on LLaVA synthetic data and original data can recognize more categories than LLaVA itself, and we also prove the insufficiency of the training model with LLaVA itself initialized compared with training on our generated data from LLaVA.
> > > >
> > > > In our experiments, we employ the equivalent parameters image encoder of BLIP and the large image encoder of LLaVA to initialize the image encoder ViT-B/16 and ViT-L/14 of DS-CLIP respectively, and other parameters are truly kept maintained with only different training data.
> > > >
> > > > As you point out, LLaVA freezes CLIP in training with the trainable linear layer, the LLaVA-initialized CLIP methods have a comparable result with ViT-rand initialized CLIP methods in our experiments as shown in the following table.
> > > >
> > > > Fine-tuning with ViT-rand initialized CLIP can achieve superior results at 78.7% compared with model training with LLaVA itself initialized, which breaks through the limitations of the LLaVA model.
> > > >
> > > > In addition, the BLIP-initialized CLIP is a fully trainable model, we also achieve an impressive accuracy of 47.7% on ImageNet zero-shot classification, surpassing the performance of BLIP-initialized CLIP methods (44.3%) by a significant margin.
> > > >
> > > > Therefore, training with a well-initized LLM-based model is insufficient compared with training with generated data from LLM.
> > > >
> > > > ###### Table 1: Ablation experiments on zero-shot ImageNet classification with different training data. All models are trained end-to-end.
> > > >
> > > > | Method  | Init. of Image Enc. | Training Data                       | Top-1 |
> > > > | ------- | ------------------- | ----------------------------------- | ----- |
> > > > | CLIP    | ViT rand.           | Laion400M                           | 75.0  |
> > > > | CLIP    | LLaVA init.         | Laion400M                           | 76.0  |
> > > > | DS-CLIP | ViT rand.           | Laion400M+generated data from LLaVA | 78.7  |
> > > > | CLIP    | BLIP init.          | --                                  | 43.8  |
> > > > | CLIP    | ViT rand.           | YFCC15M                             | 43.4  |
> > > > | CLIP    | BLIP init.          | YFCC15M                             | 44.3  |
> > > > | DS-CLIP | ViT rand.           | YFCC15M+Generated data from LLaVA   | 47.7  |
> > > >
> > > >
> > > > > Secondly, what does it mean that COCO and Flickr have a significantly smaller number of categories? Do you mean cluster? Since they do not have class information, each instance corresponds to a distinct class.
> > > >
> > > >
> > > > COCO and Flickr are proposed to verify the cross-mode alignment, COCO and Flickr focus on the complex interaction combinations of common objects.
> > > >
> > > > The visual concept has a significantly smaller number of categories compared with ImageNet.
> > > >
> > > > The number of COCO captions nouns and Flickr mapping to Wordnet are 318 and 220, respectively.
> > > >
> > > >
> > > >
> > > > > Thirdly, as the authors pointed out, the experimental results from BLIP are absent in your experiment. We cannot be sure whether their approach is effective or not.
> > > >
> > > >
> > > > The performance of BLIP on ImageNet zero-shot classification is absent in their paper, but in our experiments, we respectively evaluate the zero-shot BLIP model and utilize a pre-trained image encoder of BLIP to initialize our model on ImageNet zero-shot classification, which achieves 43.8% and 44.3% performance in Table 1.
> > > >
> > > > However, their performance is far below our method (47.7%) with only different training data and keeping the same setting of other parameters, we prove that synthetic data can improve model performance on large vocabulary recognition tasks compared with BLIP.
> > > >
> > > >
> > > >
> > > > > Finally, I'm not sure "whether synthetic data can improve model performance on large vocabulary recognition tasks, such as ImageNet zero-shot classification" is a critical question. It seems a naturally extended task of retrieval and captioning tasks.
> > > >
> > > >
> > > > COCO and Flickr focus on the complex interaction combinations of common objects, for example, most texts are similar "A woman rides a bicycle on a road next to the median.".
> > > >
> > > > ImageNet focuses on evaluating more categories of classification, such as distinguishing classes "Old English Sheepdog, Shetland Sheepdog, or German Shepherd Dog".
> > > >
> > > > In particular, we want to evaluate that if LLaVA does not know "Old English Sheepdog", training combining the generated data by LLaVA itself with the original data whether could recognize the class. It is important because we need to confirm that the recognition ability of our model is not limited by the effect of LLaVA itself.
> > > >
> > > > As shown in the experiments, our model shows superior performance on ImageNet zero-shot classification.

---

> > > > > ### Comment · Reviewer_AY1m · 2023-11-23
> > > > >
> > > > > First of all, I appreciate the time and effort you have put into addressing my concerns.
> > > > >
> > > > > The main claim of authors was that "the use of noisy captions is the problem in VLP" (which is the same claim with BLIP). It is already shown in BLIP that more descriptive (more aligned) captions are helpful.
> > > > > According to additional replies from authors, the main contribution of paper is then summarized as:
> > > > > "Instead of using less descriptive captions (original noisy captions or BLIP-generated captions), if more descriptive captions from recent LVLM (LLaVA-generated captions) is used, the recognition ability of model could be significantly increased, allowing for the representation of a wider range of categories."
> > > > >
> > > > >
> > > > > While this finding seems new, I remain cautious about the novelty and contribution about this paper since it seems straight-forward applications of LLaVA-generated captions, which are better than other variants.
> > > > > (As we all know, the captions from LLaVA, which leverages LLM, generates much more powerful captions than others. )
> > > > > Moreover, since the core ability of LLaVA lies in generating diverse captions (with frozen ViT, LLM) and authors experiment with the frozen ViT, I'm not sure it is a valid experiment to show "LLaVA synthetic data and original data can recognize more categories than LLaVA itself". I rather think it is some kind of fine-tuning step of LLaVA (exclusively for ViT) to apply for other downstream tasks like image recognition and retrieval tasks.
> > > > >
> > > > >
> > > > > To show this is a novel application (to show why descriptive captions are helpful in recognizing ability, especially in terms of encompassing a broader range of categories. ), I believe more detailed explanations and back-up experiments should be required.
> > > > >
> > > > >
> > > > > Therefore, I have chosen to maintain my reservation on this point. However, I am still open to reconsidering my stance if other reviewers find the novelty and contributions to be substantial enough for acceptance.

---

### Official Review · Reviewer_YX2M · 2023-11-02

**Soundness:** 2 fair
**Presentation:** 1 poor
**Contribution:** 1 poor
**Rating:** 3
**Confidence:** 5

**Summary:**

This paper proposed DS-CLIP for vision-language pre-training, which contains several techniques to improve original CLIP. 1) Data De-Duplication (D^3) is used for data sampling. 2) Diverse Captions Training Mechanism (DCTM) and Modality Self-enhancement Training Mechanism (MSTM) for improving the quality of the original caption. They show the proposed techniques can improve the training efficiency and final performance of zero-shot evaluation.

**Strengths:**

1. The authors conducted extensive experiments with various setups.
2. The overall performances on several benchmarks are stronger than the original CLIP.

**Weaknesses:**

1. I think the presentation is really bad and confusing. a) There are several abbreviations are introduced in the abstraction and introduction, e.g., D^3, SE, DS-CLIP, DCTM, and MSTM. Additionally, those abbreviations seem to have a hierarchical structure, DS-CLIP is for SE and D^3, SE is for DCTM and MSTM, which is really confusing. b) Some parts of the presentation are unclear. What's Image-to-Text
Multi-Positive Contrastive Loss and Text Multi-Positive Self-Supervised Loss in Fig. 2? What's the hyper-parameter choice for K, $\alpha$ and $\beta$? c) Some illustrations can be improved. In Fig. 2(a) the original image and the augmented image are reversed and the spacing between letters is different. d) Several dataset abbreviations are introduced in Sec. 4.1. However, those abbreviations are used in Sec. 4.3. You'd better define them when used. e) The ablation results in Tab. 1 are hard to read. What's your default setting and final setting for those experiments?
2. While there are several techniques are introduced in this paper, many of them are already proposed in prior arts. DCTM has been proposed in LaCLIP (Fan et al.). MSTM was introduced in DeCLIP (Li et al.). I can't find the main contribution of this paper. If those techniques are not your contribution, do not claim it. What's your main point and how does your main contribution affect the final performance?

**Questions:**

See the weakness part.

---

> ### Author Response · Authors · 2023-11-18
> **Response to YX2M (Part 1)**
>
> Thanks for your insightful advice and valuable questions, we will respond to your concerns point by point.
>
> > **Q1: Confusing about abbreviations $\text{D}^3$, SE, DS-CLIP, DCTM, and MSTM in abstract and introduction.**
>
>
> I'm sorry that these abbreviations have confused you.
>
> In this abstract and introduction, **DS-CLIP** is a simple and efficient training algorithm for multimodel representation learning.
>
> **Data De-Duplication ($\text{D}^3$)** is proposed to reduce training costs without lossing data diversity.
>
> Specifically, we uniformly sample a certain percentage of data points from each pre-clustered center, producing a small subset of the dataset. Therefore, the duplicate data were reduced according to the above re-sampling at each training epoch.
>
> **Semantic Enhancement (SE)** is proposed to address the issue of image-text misalignment and enrich textual diversity. Different from LaCLIP only with generated texts from LLM, we augment texts from the Large Language Model (LLM) and Visual Large Language Model (VLLM).
>
> With generated multiple texts, we utilize the Diverse Captions Training Mechanism (DCTM) and Modality Self-enhancement Training Mechanism (MSTM) to learn a superior representation model.
>
> **DCTM** is proposed to use all the generated captions simultaneously, this is shown to improve the performance in experiments. Concretely, it calculates the mean of the contrastive loss for the image to each corresponding text, and standard contrastive loss for the original text to the original image as shown in Eqn (1) and Eqn (2).
>
> **MSTM** leverages self-supervision within each modality for self-enhancement, the purpose is the same with DCTM. Similar to DCTM, MSTM calculates the mean of the self-supervision loss for the multiple text pairs, and standard self-supervision loss for the image and its augmented image as shown in Eqn (3) and Eqn (4).
>
> We add more explanation and remodify the logic of these representations of motivation. The revised motivation, core idea, and contributions about these abbreviations are shown in General Response.
>
> > **Q2: Unclear presentation about Image-to-Text Multi-Positive Contrastive Loss and Text Multi-Positive Self-Supervised Loss in Fig.2 and hyper-parameter choice.**
>
>
> Thank you for your suggestions.
>
> 1. The calculated processing of **Image-to-Text Multi-Positive Contrastive Loss** is as follows: one image has $M+1$ multiple texts and each image with one corresponding text calculates a standard contrastive Loss, then the mean of $M+1$ Multiple Contrastive Losses is the final loss as shown in the second one on the left in Fig.2(c).
>
>    **Text Multi-Positive Self-Supervised Loss** is as follows: $M$ pairwise text pairs in the multiple texts calculate the standard Self-Supervised Loss, and then the mean of Multiple Self-Supervised Loss is the final loss as shown in the first one on the right in Fig.2(c).
>
>    We have already modified Fig.2 and added these explanations in the revised paper.
>
> 2. The hyper-parameter of clustered number $K$ is discussed in the ablation study in Appendix.D, we explore three experiments of different clustering numbers (1000, 10000, 100000). We observed a small improvement in classification accuracy as the number of clusters increased (34.2%, 34.6%, 34.7%), which had little impact. We choose the number of clustering (10000) as default.
>
>    The hyper-parameter of $\alpha$ and $\beta$ are equal to 1 in training loss as presented in Training Settings of Appendix.B.
>
> > **Q3: Improved representation of Fig.2. e.g. the original image and the augmented image are reversed and the spacing between letters is different.**
>
> Thank you for your suggestions. The term "enhancement" in our context refers to the augmentation techniques used in self-supervised training of images. The reversed augmented image is transformed by ''FLIP'' as an example. The details of Fig.2. is modified in the revised version. We also upload the modified Fig.2 to the following link. [Figure 2](https://clip-multimodal.oss-cn-beijing.aliyuncs.com/siyang/paper/ICLR2024/figure2.png?OSSAccessKeyId=LTAI5tGRJ62Fo4Ly48B8Zd1g&amp;Expires=3601700205795&amp;Signature=4oPO76e0VqMxZThzOfCvBUeXzVg%3D)
>
>
> > **Q4: Defined Several dataset abbreviations in Sec4.3 rather than Sec4.1.**
>
> We appreciate your constructive advice. We move these dataset abbreviations from Sec4.1 into Sec4.3 in the revised paper as follows. ''Evaluations are conducted on 10 widely used visual recognition benchmarks including ImageNet (IN), ImageNetV2 (INV2), CIFAR10 (C10), CIFAR100 (C100), Caltech101 (Cal101), Oxford Pets, SUN397 (S397), Food101 (F101), DTD and Stanford Dogs.''

---

> > ### Author Response · Authors · 2023-11-18
> > **Response to YX2M (Part 2)**
> >
> > > **Q5: Explanation of the experimental setting of ablation results in Tab.1.**
> >
> >
> > Thank you for your suggestions.
> >
> > (1) We conduct extensive experiments related to our proposed $\text{D}^3$, SE, and training mechanisms DCTM and MSTM with different settings in Tab.1. The pre-trained dataset is YFCC15M for ViT-B/32 and ViT-B/16, and Laion400M is used for ViT-L/14.
> >
> > Tab.1 consists of **three parts** with the model architecture ViT-B/32, ViT-B/16, and ViT-L/14.
> >
> > **The first part** is designed to verify the effect of each component of the proposed module.
> >
> > **The second and third parts** are presented to prove that generated text from well-pre-trained VLLM is more helpful than training initialization based on the image encoder of VLLM in the aspect of the different model architecture. Specifically, the second part employs the equivalent parameters image encoder of BLIP[1] to initialize the image encoder ViT-B/16, and the third part employs the large image encoder of LLaVA to initialize the image encoder ViT-L/14.
> >
> > **In addition**, we combine three parts to prove that DS-CLIP is scalable for training on large datasets and models. We readjusted Tab.1 with clearer representations and added two experiments on ViT-L/14 aligned with Part 2.
> >
> > The following analysis is modified in Sec4.2.
> >
> > (2) In particular, the baseline is the CLIP of Experiment ID 1 in Tab.1, which uses 100% data to pre-train the CLIP without $\text{D}^3$, SE, and training mechanisms DCTM and MSTM. The default setting is DS-CLIP of Experiment ID 6 in Tab.1 sampling 50% training data according to the pre-clustered prototype with SE and training mechanism DCTM and MSTM, which is used for image classification tasks as our default setting. For other downstream tasks, we use VIT-B/16 as Experiment ID 13 in Tab.1 to initialize the image encoder of CLIP for fair comparison following the original code of MIMdet, MMSegmentation, and CoCa, and other parameters are maintained.
> >
> > The other default setting of the hyper-parameter clustered number $K$ is equal to 10000, using the image feature for clustering and uniform sampling type.
> >
> > We will change it in the revised paper.
> >
> > ###### Tab.1: Ablation experiments on zero-shot ImageNet classification with our proposed components. All 100%, Random 50%, and Clustered 50% represent training with all data, randomly selecting 50% data for training and sampling 50% data according to the pre-clustered prototype during the training epoch. "--" is no training. The default setting of hyper-parameter clustered number K is equal to 10000, using the image feature for clustering and uniform sampling type.
> >
> > | ID | Method | Init. of Image Enc. | $\text{D}^3$ | SE/LLaVA | SE/LLaMA | DCTM | MSTM | ImageNet Top-1 |
> > | --: | --- | --- | --- | --- | --- | --- | --- | --- |
> > | Dataset: YFCC15M; Model Architecture: ViT-B/32 |  |  |  |  |  |  |  |  |
> > | 1 | CLIP | ViT rand. | All 100% | ✗ | ✗ | ✗ | ✗ | 37.7 |
> > | 2 | CLIP | ViT rand. | Random 50% | ✗ | ✗ | ✗ | ✗ | 32.5 |
> > | 3 | DS-CLIP | ViT rand. | Clustered 50% | ✗ | ✗ | ✗ | ✗ | 34.6 |
> > | 4 | DS-CLIP | ViT rand. | Clustered 50% | ✓ | ✓ | ✗ | ✗ | 37.9 |
> > | 5 | DS-CLIP | ViT rand. | Clustered 50% | ✓ | ✓ | ✓ | ✗ | 38.5 |
> > | 6 | DS-CLIP | ViT rand. | Clustered 50% | ✓ | ✓ | ✓ | ✓ | 39.3 |
> > | 7 | DS-CLIP | ViT rand. | All 100% | ✓ | ✓ | ✗ | ✗ | 41.7 |
> > | 8 | DS-CLIP | ViT rand. | All 100% | ✓ | ✓ | ✓ | ✓ | **42.1** |
> > | Dataset: YFCC15M; Model Architecture: ViT-B/16 |  |  |  |  |  |  |  |  |
> > | 9 | CLIP | ViT rand. | All 100% | ✗ | ✗ | ✗ | ✗ | 43.4 |
> > | 10 | CLIP | BLIP init. | -- | ✗ | ✗ | ✗ | ✗ | 43.8 |
> > | 11 | CLIP | BLIP init. | All 100% | ✗ | ✗ | ✗ | ✗ | 44.3 |
> > | 12 | DS-CLIP | ViT rand. | All 100% | ✓ | ✗ | ✗ | ✗ | 45.7 |
> > | 13 | DS-CLIP | ViT rand. | Clustered 50% | ✓ | ✓ | ✓ | ✓ | 45.4 |
> > | 14 | DS-CLIP | ViT rand. | All 100% | ✓ | ✓ | ✓ | ✓ | **47.7** |
> > | Dataset: Laion400M; Model Architecture: ViT-L/14 |  |  |  |  |  |  |  |  |
> > | 15 | CLIP | ViT rand. | All 100% | ✗ | ✗ | ✗ | ✗ | 75.0 |
> > | 16 | CLIP | LLaVA init. | -- | ✗ | ✗ | ✗ | ✗ | 75.3 |
> > | 17 | CLIP | LLaVA init. | All 100% | ✗ | ✗ | ✗ | ✗ | 76.0 |
> > | 18 | DS-CLIP | ViT rand. | All 100% | ✓ | ✗ | ✗ | ✗ | 76.7 |
> > | 19 | DS-CLIP | ViT rand. | Clustered 50% | ✓ | ✓ | ✓ | ✓ | 76.1 |
> > | 20 | DS-CLIP | ViT rand. | All 100% | ✓ | ✓ | ✓ | ✓ | **78.7** |

---

> > > ### Author Response · Authors · 2023-11-18
> > > **Response to YX2M (Part 3)**
> > >
> > > > **Q6: DCTM and MSTM are not new. Relationship to existing methods (LaCLIP[2] and DeCLIP[1]).**
> > >
> > > **DeCLIP[1]** is proposed is to fully exploit all the self-supervised losses of augmented data, which focuses on the design of a training strategy rather than solving the problem of web data noise.
> > >
> > > **The essential difference with our method** is that we start from the misaligned between image and text, and use the rich knowledge of LLM/VLLM to enrich text representation. We focus on the problem of data noise from the source rather than the training strategy.  MSTM is proposd for the new generated data, the augmented multiple texts are naturally proper for MSTM, it truly improves the performance of representation learning. We retrained DeCLIP with our generated texts instead of data augmented. Experiments have demonstrated that using our augmented data is further elevated on DeCLIP as shown in the following Table.
> > >
> > > ###### Tab.2: Performance on 10 downstream datasets with different methods (%). DeCLIP* and LaCLIP* represent the original DeCLIP and LaCLIP, DeCLIP++ and LaCLIP++ are training with our augmented texts.
> > >
> > > | Evaluation Type | Method | IN | INV2 | Pets | C10 | C100 | S397 | F101 | Cal101 | DTD | Dogs | Avg. |
> > > | --- | --- | --- | --- | --- | --- | --- | --- | --- | --- | --- | --- | --- |
> > > | Zero-shot | DeCLIP*[1] | 43.2 | 36.1 | 30.2 | 72.1 | 39.7 | 51.6 | 46.9 | 70.1 | 24.2 | 11.7 | 42.6 |
> > > |  | DeCLIP++ | 45.2 | 38.2 | 33.0 | 72.9 | 39.9 | 53.3 | 48.4 | 72.7 | 25.3 | 11.9 | 44.1 |
> > > |  | LaCLIP*[2] | 38.1 | 34.2 | 32.1 | 77.5 | 50.3 | 47.2 | 45.8 | 79.8 | 27.5 | 10.2 | 44.3 |
> > > |  | LaCLIP++ | 38.9 | 35.2 | 33.2 | 77.9 | 50.8 | 47.6 | 46.4 | 79.9 | 28.5 | 11.3 | 44.9 |
> > > | Linear probe | DeCLIP*[1] | 69.2 | 53.1 | 76.5 | 88.6 | 71.6 | 75.9 | 79.3 | 88.0 | 69.1 | 49.9 | 72.1 |
> > > |  | DeCLIP++ | 71.3 | 54.4 | 75.9 | 87.2 | 71.9 | 76.8 | 79.9 | 88.1 | 69.8 | 51.3 | 72.7 |
> > > |  | LaCLIP*[2] | 71.1 | 54.5 | 73.1 | 91.0 | 75.6 | 73.1 | 75.8 | 89.9 | 73.8 | 53.4 | 73.1 |
> > > |  | LaCLIP++ | 72.0 | 55.3 | 73.9 | 91.6 | 75.9 | 73.8 | 76.3 | 89.7 | 74.7 | 53.9 | 73.7 |
> > >
> > > **LaCLIP[2]** rewrites and changes the structure of text from the original text using LLaMA and chatbots but maintains semantic information of the text. Text rewritten in LaCLIP only ensures the semantics associated with the corresponding original text but it does not address the misalignment problem of image and text.
> > >
> > > **The advantage of DS-CLIP is that** we can supplement the shortcomings of LaCLIP from the aspects of text augmented via VLLM. DCTM is a training strategy for enriching text in both LaCLIP and our method. The augmented multiple texts are naturally proper for DCTM. We combine our generated data with LaCLIP, and the average mAP on 10 image classification datasets is improved with our augmented data.
> > >
> > > **Reference:**
> > >
> > > [1] Li Y, Liang F, Zhao L, et al. Supervision exists everywhere: A data efficient contrastive language-image pre-training paradigm[J]. arXiv preprint arXiv:2110.05208, 2021.
> > >
> > > [2] Fan L, Krishnan D, Isola P, et al. Improving CLIP Training with Language Rewrites[J]. arXiv preprint arXiv:2305.20088, 2023.
> > >
> > > > **Q7: The main contribution of this paper and the effect of the main contribution on final performance.**
> > >
> > > (1) Multimodal datasets are a critical component in recent breakthroughs such as CLIP and GPT4, but their design does not receive the same research attention as model architectures or training algorithms.
> > >
> > > In this paper, we **focus on** constructing semantically diverse multimodal datasets and achieving the first publicly available high-quality image-text dataset with diverse semantic captions . In particular, we utilize LLM and VLLM to augment a more diverse dataset. Although the LLM-generated data is presented in LaCLIP, it overlooks the importance of the visual LLM(VLLM) and still exists unmatched image-text pair in the organized data.
> > >
> > > Additionally, we provide evidence to **support** that training on our newly constructed dataset leads to a model that surpassed the original VLLM model, further confirming the importance of LLM and VLLM in data construction. Please refer to the experiment 16 and 17 in Table 1 for details.
> > >
> > > The well-processed dataset will be released for further research. The motivation, idea, and contribution of DS-CLIP are re-summarised. Please refer to General Response.
> > >
> > > (2) Each proposed module including $\text{D}^3$, SE, and training mechanism DCTM and MSTM has important effects. As presented in Sec4.1, when employing a 50% sampling ratio, the model trained with pre-clustered prototype sampling in **$\text{D}^3$** has a higher 2.1% compared with the model trained with random sampling. Combining **SE** with CLIP (Experiment ID 7) during training improves by 4% compared to baseline CLIP (Experiment ID 1). With training strategy **DCTM** and **MSTM**, the performance of DS-CLIP further improves by 1.4% (Experiment ID 4 vs Experiment ID 6). More analyses are presented in Sec4.1 in the revised paper.

---

> > > > ### Comment · Reviewer_YX2M · 2023-11-22
> > > >
> > > > Thanks for the reply regarding my concerns. I appreciate the detailed reply. I have carefully read them and choose to keep my original score.

---

### Author Response · Authors · 2023-11-16
**General Response**

We thank all reviewers for their dedication to our paper and insightful comments, and we believe these comments are significant for improving the overall quality of this paper.

We are pleased that the reviewers appreciate our paper from various aspects, including its stronger performance [YX2M] [xVnC] [AY1m] [Fx4U], detailed experiments [xVnC] [Fx4U] [YX2M], well-written and easy-to-follow [AY1m] [Fx4U] [xVnC], solid motivation [AY1m], efficient method [Fx4U]. These positive assessments truly motivate us.

The **motivation** of DS-CLIP is to solve the following problem:

- Multimodal datasets are critical in recent breakthroughs such as CLIP and GPT4, but their design does not receive the same research attention as model architectures or training algorithms.

  The web data are noisy, with significant scene redundancy and misalignment in the image-text pairs, model training on these data is expensive and requires more computing resources.

Thus, the **core idea** of DS-CLIP is to construct a semantically diverse multimodal dataset and present an efficient training algorithm for better representation learning, we will release the dataset for future research.

To effectively utilize the semantically diverse multimodal dataset, we conduct the clustering of data and uniformly sample half of the data from each pre-clustered center at each training epoch, further using the Diverse Captions Training Mechanism (DCTM) and Modality Self-enhancement Training Mechanism (MSTM) achieve the superior performance of CLIP. Training with less data on a semantically diverse dataset can achieve superior performance and save computing resources, which is important for studies with limited resources in the laboratory. Additionally, we provide evidence to support that training on our newly constructed dataset leads to a model that surpasses the original VLLM model, further confirming the importance of LLM and VLLM in data construction. Please refer to the experiments 16 and 17 in Table 1 for details.

We summarize our **contributions** as follows:

-  We construct a semantically diverse multimodal dataset (the first publicly available high-quality image-text dataset with diverse semantic captions) for better representation learning through the Semantic Enhancement (SE) module, which effectively addresses the issue of image-text misalignment and enriches textual diversity.
-  To effectively utilize the semantically diverse multimodal dataset, we design the Data De-Duplication ($\text{D}^3$), which manages a paradigm to reduce training costs without losing data diversity.
-  With SE and $\text{D}^3$, we present an efficient training algorithm, called DS-CLIP using the Diverse Captions Training Mechanism (DCTM) and Modality Self-enhancement Training Mechanism (MSTM) to learn a superior representation model. Our proposed method significantly outperforms CLIP on special fine-grained classification datasets and various patch-level downstream tasks from 0.2% to 23.9%, with ONLY half of the training time.

---

### Meta-Review · Area_Chair_gKmD · 2023-12-06

**Metareview:**

**Summary**

This submission proposes DS-CLIP for vision-language pre-training. The key technical components are: 1) Data De-duplication (D3) via pre-clustered, 2) Semantic Enhancement (SE) by generating more descriptive captions with powerful large pre-trained models, and 3) Diverse Captions Training Mechanism (DCTM) and a Modality Self-enhancement Training Mechanism (MSTM) to utilize diverse captions and to combine uni-modal contrastive learning.

**Strengths**

- All reviewers think the submission has conducted extensive experiments and some mentioned the empirical performance is strong, both in accuracy and efficiency.
- Most reviewers think this paper easy well-written and easy to follow. A reviewer proposed several presentation improvement ideas, and the authors has clarified in the rebuttal. Following the suggestion, I believe the authors can improve the manuscript to be more inclusive and easy to read for our diverse community.

**Weaknesses**
- All reviewers share similar novelty concerns -- The high-level idea of generating more captions to improve VL pre-training is not new. Several reviewers point out that it's basically the same core contribution of BLIP, but with minor modification / stronger, newer pre-trained models.
- Reviewers point out that other findings such as diversity of captions helps and the unimodal contrastive learning's benefits are known from prior work. Reviewers are not impressed by the finding.
- Some reviewers suggested additional experiments and writing improvements. The authors have addressed them, at least to some extent during the rebuttal. However, reviewers did not think the improvements offset the lack of novelty.

**Justification For Why Not Higher Score:**

It seems all reviewers share a common concern on lack of novelty. Generate captions for VL pre-training is not new at all, and it has become a common belief that higher data (caption) quality leads to stronger pre-trained model. Reviewers are not impressed on the use of newer, stronger pre-trained (V)LLMs to achieve stronger empirical results, and I share the some feeling as the area chair. The technical contribution seems a naive extension of BLIP with modern LLM, hence the reviewers and I all agree that this submission does not meeting the bar for ICLR.

Some additional comments:

- Reviewer YX2M's engagement in discussion wasn't active enough, so I've downweighed his/her review. Still, YX2M's concerns are valid, and other reviewers share similar concerns.
- I personally like the direction of improving data quality, rather than some complex modeling / algorithmic improvements that are not scalable. Unfortunately, I did not find impressive technical contribution enough to overrule the collective assessment of the reviewers. To encourage the authors to improve along the direction, I suggest them to maybe include generative VL application such as VLLM and diffusion. After all, the community seems not surprised by enhancement about CLIP-like contrastive models anymore. Generative models, on the other hand, could be more impactful in the future.

**Justification For Why Not Lower Score:**

N/A

---

### Decision · Program_Chairs · 2024-01-16

Reject